# Unified control of amoeboid pseudopod extension in multiple organisms by branched F-actin in the front and parallel F-actin/myosin in the cortex

Peter J. M. van Haastert ⬥ *

Department of Cell Biochemistry, University of Groningen, Groningen, The Netherlands

* p.j.m.van.haastert@rug.nl

**Data Availability Statement:** All relevant data are within the manuscript and its Supporting Information files.

## Abstract

The trajectory of moving eukaryotic cells depends on the kinetics and direction of extending pseudopods. The direction of pseudopods has been well studied to unravel mechanisms for chemotaxis, wound healing and inflammation. However, the kinetics of pseudopod extension–when and why do pseudopods start and stop- is equally important, but is largely unknown. Here the START and STOP of about 4000 pseudopods was determined in four different species, at four conditions and in nine mutants (fast amoeboids *Dictyostelium* and neutrophils, slow mesenchymal stem cells, and fungus *B.d. chytrid* with pseudopod and a flagellum). The START of a first pseudopod is a random event with a probability that is species-specific (23%/s for neutrophils). In all species and conditions, the START of a second pseudopod is strongly inhibited by the extending first pseudopod, which depends on parallel filamentous actin/myosin in the cell cortex. Pseudopods extend at a constant rate by polymerization of branched F-actin at the pseudopod tip, which requires the Scar complex. The STOP of pseudopod extension is induced by multiple inhibitory processes that evolve during pseudopod extension and mainly depend on the increasing size of the pseudopod. Surprisingly, no differences in pseudopod kinetics are detectable between polarized, unpolarized or chemotactic cells, and also not between different species except for small differences in numerical values. This suggests that the analysis has uncovered the fundament of cell movement with distinct roles for stimulatory branched F-actin in the protrusion and inhibitory parallel F-actin in the contractile cortex.

## Introduction

Many eukaryotic cells move by making protrusion [1]. Upon flow of cytoplasm into the protrusion, the center of mass of the cell displaces and the cell has effectively moved in the direction of the extending protrusion. These protrusions can be long-lived as in keratocytes, which glide with a single broad anterior protrusion that is continuously extending and filled with cytoplasm. However, in most cells the protrusions are transient with a short phase of extension and filling with cytoplasm, followed by the formation of a new protrusion [1]. In amoeboid

**Funding:** The author(s) received no specific funding for this work.

**Competing interests:** The authors have declared that no competing interests exist.

cells, such as neutrophils and *Dictyostelium*, these protrusions are in the form of pseudopods that extend fast at a high frequency; neutrophils rapidly move to areas with infections at a rate of about 10 μm/min [2]. In mesenchymal-type cells such as fibroblast, the protrusions are in the form of filopodia and lamellipodia that extend much slower but with strong attachment to the substratum; fibroblast slowly close a wound at a rate of about 0.5 μm/min [3,4].

Pseudopods are extended perpendicular to the cell surface [5]. Therefore, the path of a moving cell depends on the time and place where a series of pseudopods are formed [6–10]. Unpolarized cells start new protrusions at nearly random positions, resulting in near Brownian motion [6,11]. In polarized cells new pseudopods are preferentially formed at a stable front close to previous pseudopods, and inhibited at the side and in the rear of the cell, resulting in movement with strong persistence, but not in a specific direction [6,8,11,12]. During chemotaxis, pseudopods are preferentially made at the side of the cell facing the highest concentration of chemoattractant, and cells move with persistence and direction [9,13–15]. Several signaling pathways have been elucidated regulating the place of pseudopods formation [13,16–19]. Myosin filaments at the side and in the rear inhibit pseudopod formation, while Arp2/3-Scar induced F-actin in the front stimulates pseudopod formation [1,10,20–24]. These signaling pathways are regulated by internal factors in polarized cells and by external factors during chemotaxis.

In contrast to the many studies on the position of pseudopod extension, the timing of pseudopod extension is less frequently studied [7,9,25]. Why and how does a pseudopod starts its extension? Why and how does it stop its extension? Why have most cells only one extending pseudopod? Is the kinetics of pseudopod extension similar or very different in polarized versus non-polarized cells, in neutrophils versus fibroblasts, in cells in buffer versus chemotaxing cells? To address these largely unanswered questions, the time and position at the start and stop of pseudopods extension was determined. Data were collected for about 4000 pseudopods from *Dictyostelium* at four conditions (unpolarized, polarized, chemotaxis and under agar), nine *Dictyostelium* mutants with deletion of specific components or regulators of the cytoskeleton, and four species (the fast amoeboids *Dictyostelium* and neutrophils, the slow mesenchymal stem cells, and the fungus *B.d. chytrid* that has a pseudopod and a flagellum). Kinetic constants were derived for the regulation of the START and STOP of pseudopod extension. Unexpectedly, the data reveal very similar mechanisms of pseudopod START and STOP kinetics for all these conditions and species, which suggest that the fundament of cell movement may have been captured: The START of a first pseudopod is a random stochastic event with a probability that is species-specific. Pseudopods extension is mediated by polymerization of branched F-actin at the tip of the pseudopod. The START of a second pseudopod is strongly inhibited by the extending first pseudopod; this inhibition depends on the parallel filamentous actin/myosin in the cortex of the cell. The STOP of pseudopods extension is due to inhibition that depend largely on the pseudopod size and partly on pseudopod growth time and rate of extension. Pseudopods stops prematurely in scar-mutants with reduced branched F-actin polymerization or at conditions with increased resistance such as cells moving under agar. The data are discussed in a conceptual framework with distinct roles for stimulatory branched F-actin in the protrusion and inhibitory parallel F-actin in the contractile cortex.

## Results

### Pseudopod extension

To identify active pseudopods, the tip of extending pseudopods were followed at high temporal and spatial resolution (Fig 1A). Fig 1B reveals that during the life of pseudopods the rate of extension is approximately constant and does not involve changes in rate at the beginning or

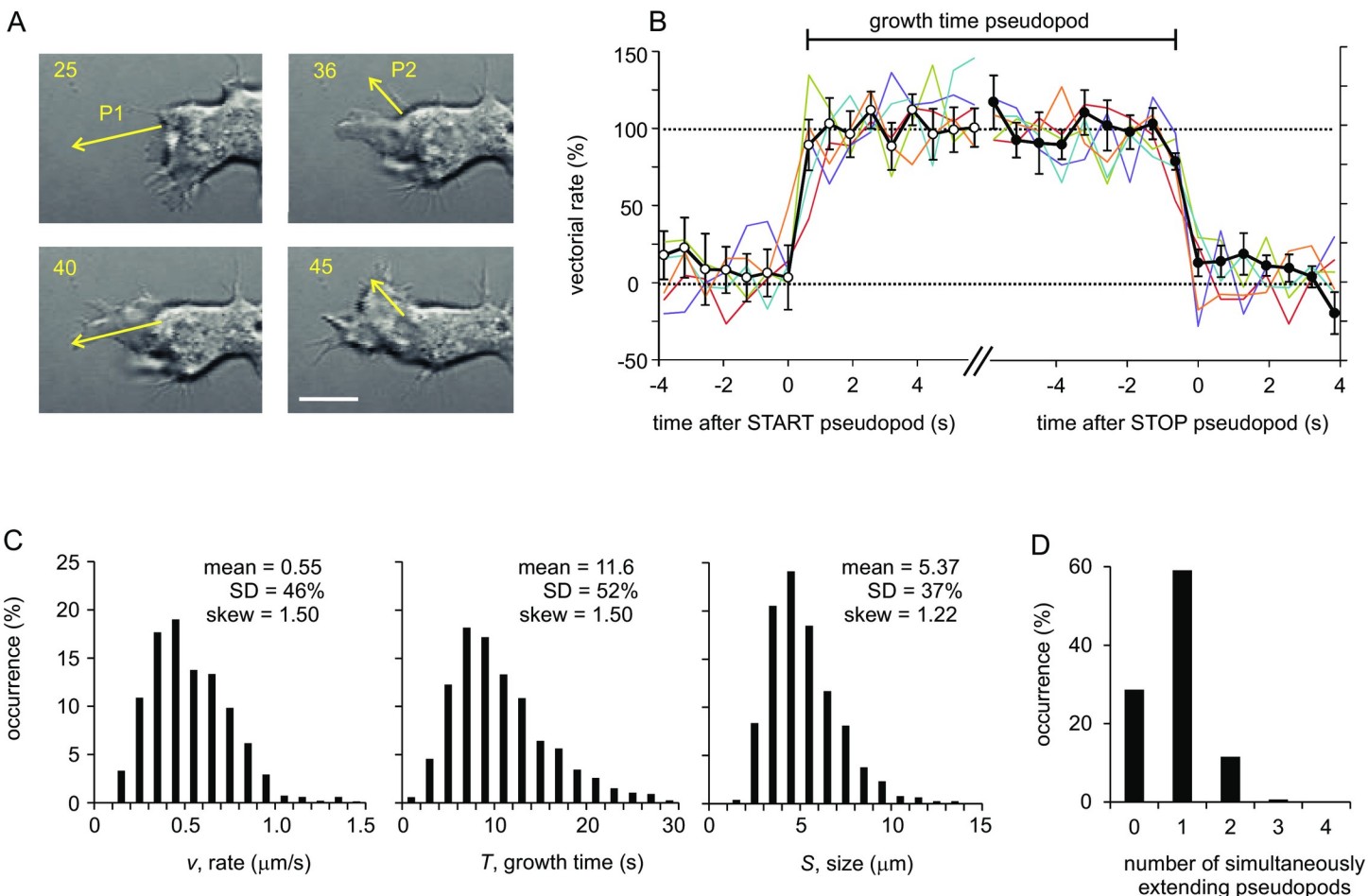

**Fig 1. Basal pseudopod properties of polarized *Dictyostelium* cells.** (A) Images of wild-type AX3 cells with frame number (1 s per frame, 245 nm pixel size) and two extending pseudopods. P1 starts at frame 25, P2 starts at frame 36, P1 stops at frame 40 and P2 stops at frame 45. The arrow connects the tip of a pseudopod at start and stop, respectively. The bar is 5 μm. (B) The rate of the tip of 10 pseudopods in the direction of the arrow at different times before, during and after extension was recorded at higher temporal and spatial resolution (0.64 s per frame, 99 nm pixel size). The rate is presented as % of the average rate during extension, which varied between 0.37 and 0.66 μm/s for the 10 pseudopods. Data during START were aligned in time for the last time moment that the speed is below 50% (open circles), and data during STOP were aligned for the first time moment that the speed is below 50% (closed circles). Date in color are five individual pseudopods; data in black are means and SEM, with n = 10 pseudopods. (C) Histograms of the rates, growth times and sizes of 996 pseudopods. The size has the smallest variation and skew. (D) Histogram of time that cells extend simultaneously 0, 1, 2, 3 or 4 pseudopods; the total time is 15,356 seconds.

towards the end of the life of pseudopods. This observation confirms previous experiments with much lower resolution [7]. Pseudopods start and stop suddenly, and switch between basal and full extension within 0.64 seconds, the time resolution of this experiment. Therefore, the kinetic process of pseudopod extension is a binary on/off switch, with stochastic or regulated probabilities to START (switch on) or STOP (switch off). To characterize the quantitative properties of these on/off switches and their molecular mechanisms, the position and time of the tip of the pseudopod was identified at its START and STOP, respectively. Data were collected for 996 pseudopods of starved wild-type *Dictyostelium* cells, and for about 100 to 200 pseudopods each for three environmental conditions, nine different mutants, and four cell type/species (all data are presented in supplemental S1 Table, and summarized in Table 1).

The 996 pseudopods of starved wild-type *Dictyostelium* cells were characterized in detail. They have a mean growth time of 11.6 seconds, extend at a rate of 0.55 μm/s and

**Table 1. Pseudopod dynamics.**

| strain or | symbol | Basic properties | | | | Start | | Stop | | | |
|---|---|---|---|---|---|---|---|---|---|---|---|
| | | size | growth time | rate | extending pseudopods | start probability | Inhibition start by pseudopod | stop probability | by rate | by size | by time |
| strain or | symbol | $S$ | $T$ | $v$ | | $\alpha$ | $A$ | $\tilde{\beta}$ | $k_v$ | $k_s$ | $k_t$ |
| condition | unit | μm | s | μm/s | number/cell | %/s | fold | %/s | s/μm | 1/μm | 1/s |
| *Dictyostelium* | n | | | | | | | | | | |
| polarized | 996 | 5.4 ±0.13 | 11.6±0.37 | 0.55 ±0.018 | 0.85±0.01 | 14.0±0.5 | 3.57±0.30 | 8.0±0.40 | 0.085 ±0.033 | 0.132 ±0.007 | 0.023 ±0.004 |
| unpolarized | 344 | 4.9 ±0.18 | 11.1±0.61 | 0.53 ±0.028 | 0.77±0.02 | 14.2±0.5 | 4.39±0.19 | 8.4±0.82 | 0.011 ±0.095 | 0.153 ±0.010 | 0.016 ±0.006 |
| chemotaxis | 414 | 5.4 ±0.20 | 11.4±0.58 | 0.55 ±0.027 | 0.80±0.02 | 16.7±0.5 | 4.06±0.88 | 8.5±0.84 | 0.100 ±0.072 | 0.139 ±0.009 | 0.018 ±0.005 |
| under agar | 218 | 4.6 ±0.27 | 7.6±0.59 | 0.71 ±0.043 | 0.95±0.03 | 33.5±3.5 | 3.69±0.42 | 13.0±1.20 | 0.427 ±0.075 | 0.120 ±0.013 | 0.022 ±0.010 |
| *mutants* | | | | | | | | | | | |
| *scar*-null | 251 | 1.4 ±0.06 | 2.6±0.18 | 0.79 ±0.045 | 0.62±0.03 | 36.3±4.7 | 3.99±0.26 | 40.9±4.85 | 0.100 ±0.030 | 0.595 ±0.040 | 0.040 ±0.003 |
| *pla2*-null | 194 | 8.2±2.9 | 19.3±1.67 | 0.54 ±0.042 | 0.78±0.02 | 11.1±1.3 | 11.84±1.06 | 5.0±0.48 | 0.110 ±0.045 | 0.085 ±0.007 | 0.013 ±0.003 |
| *gc*-null | 152 | 4.6 ±0.23 | 10.7±0.76 | 0.53 ±0.068 | 0.95±0.03 | 24.5±1.5 | 3.71±0.43 | 9.3±0.58 | 0.011 ±0.045 | 0.165 ±0.011 | 0.021 ±0.006 |
| *gbpC*-null | 163 | 5.0 ±0.27 | 9.3±0.74 | 0.64 ±0.045 | 0.98±0.04 | 27.0±2.8 | 3.66±0.57 | 10.7±0.90 | 0.016 ±0.061 | 0.160 ±0.017 | 0.016 ±0.011 |
| *myoII*-null | 170 | 5.0 ±0.22 | 13.6±0.75 | 0.42 ±0.032 | 0.93±0.03 | 12.1±1.2 | 1.82±0.52 | 7.3±0.86 | 0.002 ±0.100 | 0.128 ±0.017 | 0.026 ±0.007 |
| *Rap1G12V* | 150 | 3.1 ±0.16 | 10.4±0.78 | 0.35 ±0.029 | 1.54±0.10 | 20.0±1.4 | 1.49±0.24 | 8.0±1.17 | 0.388 ±0170 | 0.173 ±0.025 | 0.032 ±0.008 |
| *forAEH*-null | 106 | 3.5 ±0.23 | 13.4±0.94 | 0.28 ±0.018 | 1.93±0.08 | 21.0±1.7 | 1.39±0.13 | 6.7±0.64 | 0.270 ±0.090 | 0.164 ±0.021 | 0.024 ±0.005 |
| *racE*-null | 140 | 4.2 ±0.24 | 11.2±0.94 | 0.43 ±0.031 | 2.16±0.12 | 23.1±3.4 | 1.20±0.16 | 8.6±0.93 | 0.320 ±0.100 | 0.189 ±0.017 | 0.016 ±0.003 |
| *lrrA*-null | 178 | 4.3 ±0.26 | 7.9±0.70 | 0.72 ±0.092 | 1.11±0.05 | 26.6±4.4 | 1.19±0.15 | 14.5±1.37 | 0.140 ±0.059 | 0.157 ±0.015 | 0.027 ±0.012 |
| *Other species* | | | | | | | | | | | |
| Neutrophils | 150 | 3.3 ±0.21 | 5.1±0.45 | 0.75 ±0.058 | 0.66±0.03 | 22.9±2.6 | 4.20±0.39 | 18.3±1.23 | 0.078 ±0.042 | 0.250 ±0.015 | 0.023 ±0.012 |
| Mesenchymal cells | 140 | 13.8 ±0.98 | 461±70 | 0.047 ±.0056 | 0.38±0.03 | 0.128±0.013 | 4.23±0.27 | 0.255±0.070 | 0.020± 0.190 | 0.070± 0.002 | 0.00001 ±0.0001 |
| *B.d. chytrid* | 138 | 2.2 ±0.13 | 5.1±0.44 | 0.51 ±0.04 | 0.75±0.05 | 40.0±2.3 | 5.76±0.71 | 19.9±0.73 | 0.08 ±0.024 | 0.319 ±0.012 | 0.060 ±0.012 |

The data shown are the means and 95% confidence interval with n the number of pseudopods analyzed. The values for A and $\tilde{\beta}$ are the means of multiple analysis methods (see S2 Table for original data). For *Dictyostelium* conditions and mutants: Values in green are both substantially (at least 1.5 fold) and significantly (at P<0.001) different from polarized wild-type. For all strains and conditions: Values in red are not significantly different from zero (at P>0.1).

have a size of 5.4 μm; on average a cell extends about 4 new pseudopods per minute, yielding a pseudopod interval of 15.4 seconds. Histograms of these pseudopod properties are shown in Fig 1C, revealing a rather broad and slightly skewed distribution of growth times, sizes and rates. Cells may have no, one or multiple extending pseudopods (Fig 1D), which will be used later to investigate how simultaneously extending pseudopod influence each other.

A cell may extend pseudopods according to the general scheme below:

$$C_0 \underset{\beta_1}{\overset{\alpha_1}{\rightleftarrows}} C_1 \underset{2\beta_2}{\overset{\alpha_2}{\rightleftarrows}} C_2 \underset{3\beta_3}{\overset{\alpha_3}{\rightleftarrows}} C_3 \rightleftarrows \underset{n\beta_n}{\overset{\alpha_n}{\rightleftarrows}} C_n \rightleftarrows \qquad \text{Scheme 1}$$

Here $C_n$ denotes a cell with $n$ pseudopods, $\alpha_n$ is the probability to start the $n^{th}$ pseudopod and $\beta_n$ is the probability of stopping of the $n^{th}$ pseudopod (please note that a cell with $n$ pseudopods has $n$ possibilities to stop one of its pseudopod explaining the rate $n\beta_n$ of conversion of $C_n$ to $C_{n-1}$). The simplest model assumes that pseudopod START and STOP are pure stochastic processes; stochastic means that all pseudopods have the same probability to start or stop, independent of the presence of other pseudopods ($\alpha_n = \alpha$ and $\beta_n = \beta$, independent of $n$), and that these probabilities are constant and do not depend on e.g. the growth time or size of the pseudopod. Experiments described below reveal that START and STOP are not stochastic, but for very different reasons: START is strongly inhibited by the presence of other pseudopods, while STOP of a pseudopod is not influenced by other pseudopods, but is a complex function of growth time, size and rate of the extending pseudopod itself. The theoretical background for pseudopod kinetics is described in S1 Text in S1 File.

## The START of a new pseudopod is inhibited by the current pseudopod

The large data set of 996 pseudopods contains 622 cases where at the moment of pseudopod STOP the cell has no other extending pseudopod. These "naive" cells were used to investigate the START probability $\alpha_1$ of the first pseudopod. Fig 2A shows that "naive" cells rapidly start new pseudopods, after 1 second already 84 cells have started a new pseudopod, half of the cells have started a new pseudopod after about 4 seconds and nearly all 622 cells have started a new pseudopod after 30 seconds. The probability to start a pseudopod ($P_{START}$ in fraction/s) is the number of cells that start a pseudopod in a time interval of 1 second divided by the number of cells that have not yet started a pseudopod. The inset of Fig 2A shows that $P_{START}$ is nearly constant. Kinetic analysis (Fig 2B) reveals that START of the first pseudopod is a stochastic first order process (straight line) that begins immediately after stop of the previous pseudopod ($t_0 = -0.17 \pm 0.27$ s; intercept with the time-axis and 95% confidence intervals of the linear regression with n = 14 time points; $t_0$ not significantly different from 0 s), and START occurs at a rate of $\alpha_1 = 0.140 \pm 0.005$ s$^{-1}$ (slope). This implies that cells without pseudopods have a stochastic probability of 14% per second to start a new pseudopod. Next, while these 622 cells are extending their first pseudopod, the probability to start a second pseudopod was recorded. The time interval between the start of the first pseudopod and the start of the second pseudopod is presented as a kinetic plot (squares in Fig 2B). The results reveal that the start of a second pseudopod -while the first pseudopod is still extending- is also a first order process, but with much smaller rate constant of $\alpha_2 = 0.040 \pm 0.001$ s$^{-1}$. Thus, the start of a second pseudopod is inhibited about 3.5-fold by the first pseudopod; this fold-inhibition of START of the $n^{th}$ pseudopod is symbolized as $A_n$ ($A_1 = \alpha_1/\alpha_2 = 3.46 \pm 0.14$). Several cells extend for some period of time simultaneously two pseudopods (n = 277). For this population the start of a third pseudopod was measured, which again shows first order kinetics with a still lower rate constant of $\alpha_3 = 0.011 \pm 0.001$s$^{-1}$. Therefore, the start of a third pseudopod is inhibited about 13-fold by the two pseudopods ($A_2 = \alpha_1/\alpha_3 = 12.8 \pm 1.2$). Since 12.8 is approximately equal to the square of 3.5, the data may suggest that each extending pseudopod contributes to the inhibition of the start of a new pseudopod by a factor $A$. This hypothesis was tested in a meta-analysis of all 16 cell lines with different values of $A_1$, revealing that indeed the inhibition $A$ of START depends on the power of the number of extending pseudopods (S1 Fig in S1 File), i.e. $A_n = A^n$. Therefore, in the general scheme 1 the START of the $n^{th}$ pseudopod is described by only two

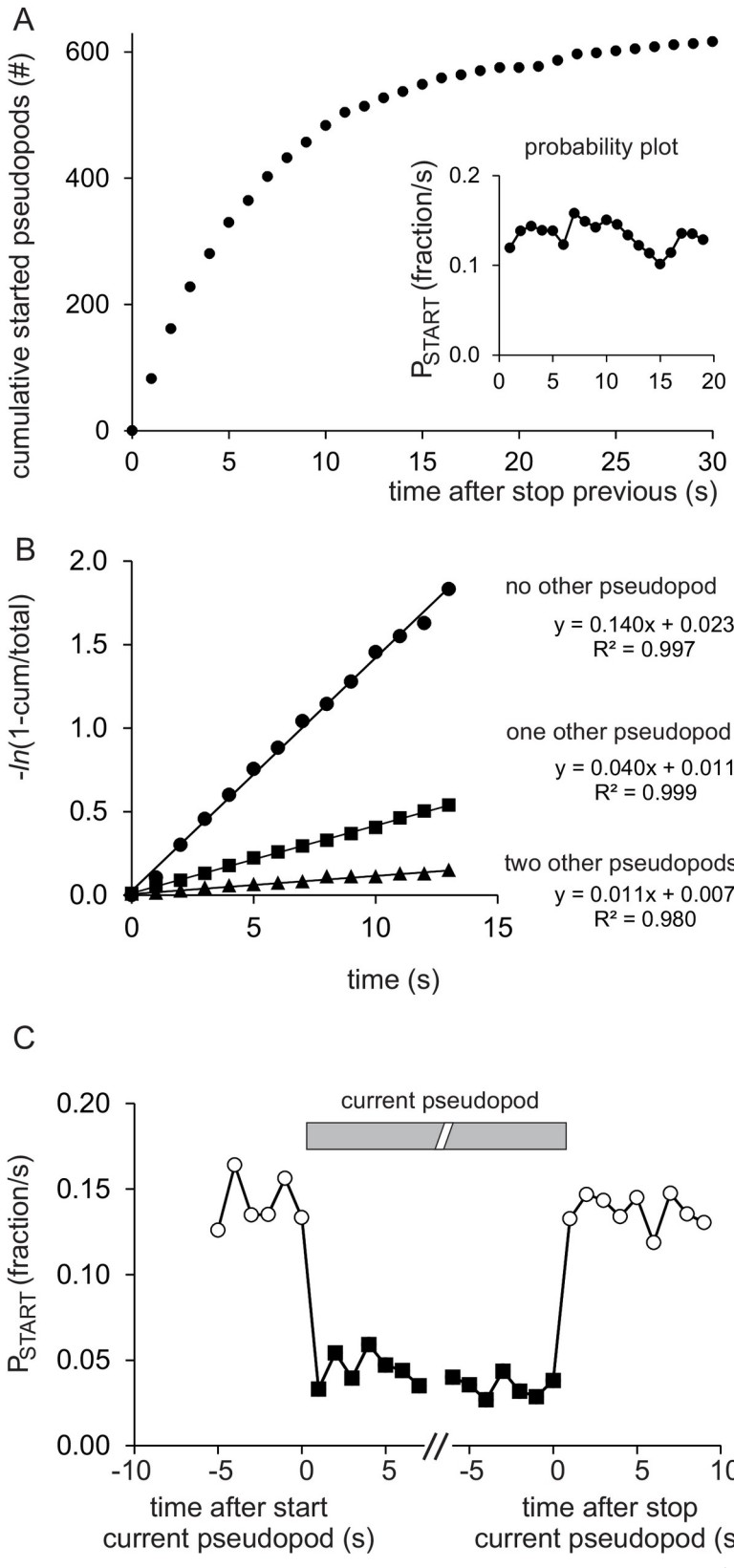

**Fig 2. The START of pseudopods is inhibited by other pseudopods.** In 622 cases out of 996 pseudopods from polarized cells, no other pseudopod was present at the moment these pseudopods stop. Investigated was how these

"naive" cells start new pseudopods. (A) Cumulative number of started first pseudopods; the inset shows the probability to start a new pseudopod ($P_{START}$). (B) Kinetic plot, analyzing the start of the first pseudopod (closed circles); time is seconds after stop of the previous pseudopod, cum is the cumulative number of cells that have started a new pseudopod at time $t$ and total = 622. Data were fitted by linear regression with n = 14 time points, yielding and intercept with the time-axis of -0.17 ± 0.27 s and a rate of 0.140 ± 0.005 s$^{-1}$ (optimal value and 95% confidence interval of the linear fit). The squared symbols show the kinetics at which these 622 cells with a first extending pseudopod will extend a second pseudopod (217 cases); time is seconds after start of the first pseudopod, cum is the cumulative number of cells that have started a second pseudopod and total is the number of the 622 cells that are still extending the first pseudopod at the time indicated. Linear regression yields an intercept with the time-axis of -0.28 ± 0.17 s and a rate of 0.040 ± 0.007 s$^{-1}$. The triangle symbols show the kinetics at which these 217 cells with two extending pseudopod will extend a third pseudopod (31 cases); time is seconds after start of the second pseudopod, cum is the cumulative number of cells that have started a third pseudopod and total is the number of the 217 cells that are still extending the first and the second pseudopod at the time indicated. Linear regression yields an intercept with the time-axis of -0.66 ± 0.69 s and a still lower rate of 0.011 ± 0.001 s$^{-1}$. The intercepts with the time-axes are statistically not significantly different from zero (t-test, P>0.1). (C) The probability to start a new pseudopod ($P_{START}$) was calculated for the group of cells before they extend a pseudopod, during the period that they extend one pseudopod, and after they have extended a pseudopod. The results reveal that the probability is always ~0.14/s for cells having no other pseudopod (open circles), and always ~0.05/s for cells with one extending pseudopod, indicating that the low probability of START of a second pseudopod appears and disappears virtually immediately. See also S2 Fig in S1 File for additional experiments supporting this conclusion.

parameters, $\alpha_1$ and $A$, according to $P_{START} = \alpha_n = \alpha_1/A^{n-1}$. For polarized wild-type cells the best estimate is $A$ = 3.59 ± 0.28 (Table 1 and S2 Table).

How fast does an extending pseudopod inhibit the START of a second pseudopod? Fig 2C reveals that within one second after a pseudopod starts, the probability to start a second pseudopod has declined to about 4%/s, and within one second after this first pseudopod stops (and cells have thus no pseudopod), the probability to start a new first pseudopod has increased to 14%/s. These results suggest that a hypothetical inhibitor leading to the 3.5-fold inhibition of the start of a new pseudopod appears and disappears within 1 second after the start and stop of the current pseudopod, respectively, which is the time resolution of the experiment. Occasionally two extending pseudopods stop nearly simultaneously; also in these cases the very low start probability of a new pseudopod during extension of the two pseudopods (1%/s) increases to 14%/s within 1 second after stop of the two pseudopods (S2 Fig in S1 File).

In summary, the START of a pseudopod is a stochastic event, but with a rate constant that is A-fold lower for each extending pseudopod present. The chemical or physical mechanism by which an extending pseudopod inhibits the start of other pseudopods appears and disappears virtually instantaneous with the presence of these other pseudopods.

## Pseudopods STOP by local inhibition

After a period of extension, pseudopods suddenly stop. To investigate the role of other pseudopods in this stopping process, the growth time and size of a pseudopod was measured for cells with different number of extending pseudopods (Fig 3A). The results reveal that the growth times and sizes of a pseudopod is not significantly different when a cell has no, one or two other extending pseudopods. Wild-type cells have maximally three extending pseudopods; however, several mutants may possess up to 6 extending pseudopods. A meta-analysis of all 16 strains reveals that the growth time and size of a pseudopod is essentially independent of other extending pseudopods present (Fig 3B). This suggest that potential inhibitory processes that builds up during the life of a pseudopod only contribute to its own stop, but do not influence the stopping of other pseudopods, if present. This mean for scheme 1 that $\beta$ is independent of $n$, or $\beta_n = \beta$.

How local is pseudopod STOP? Previously we have shown that in polarized cells the majority of new pseudopods start at the side of the previous pseudopod [13]. In those experiments

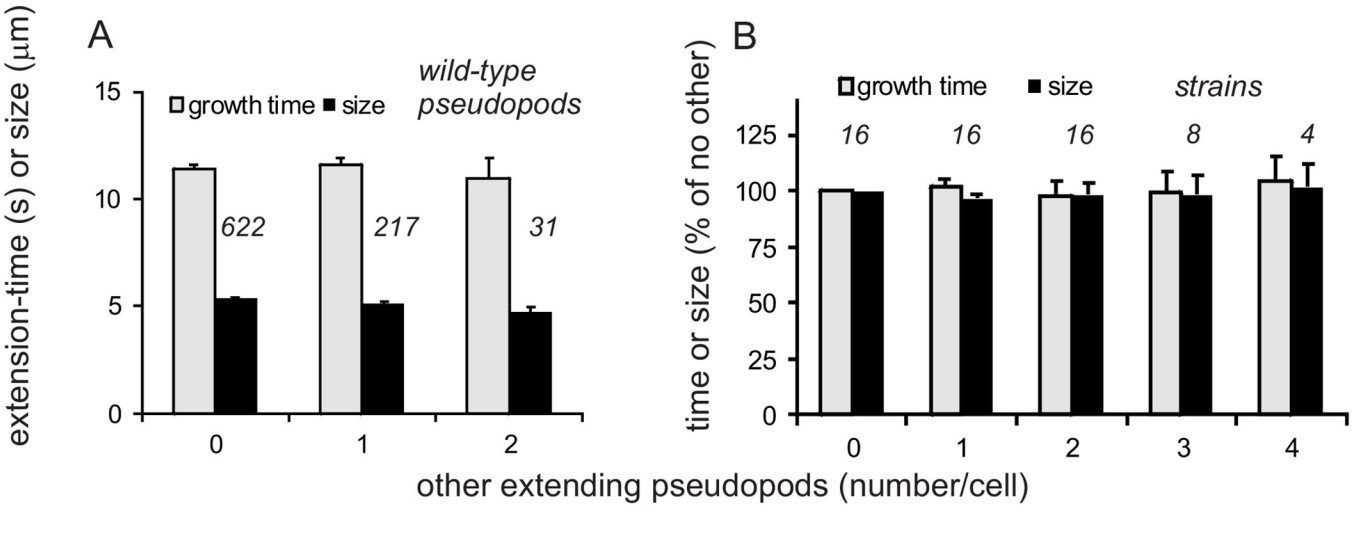

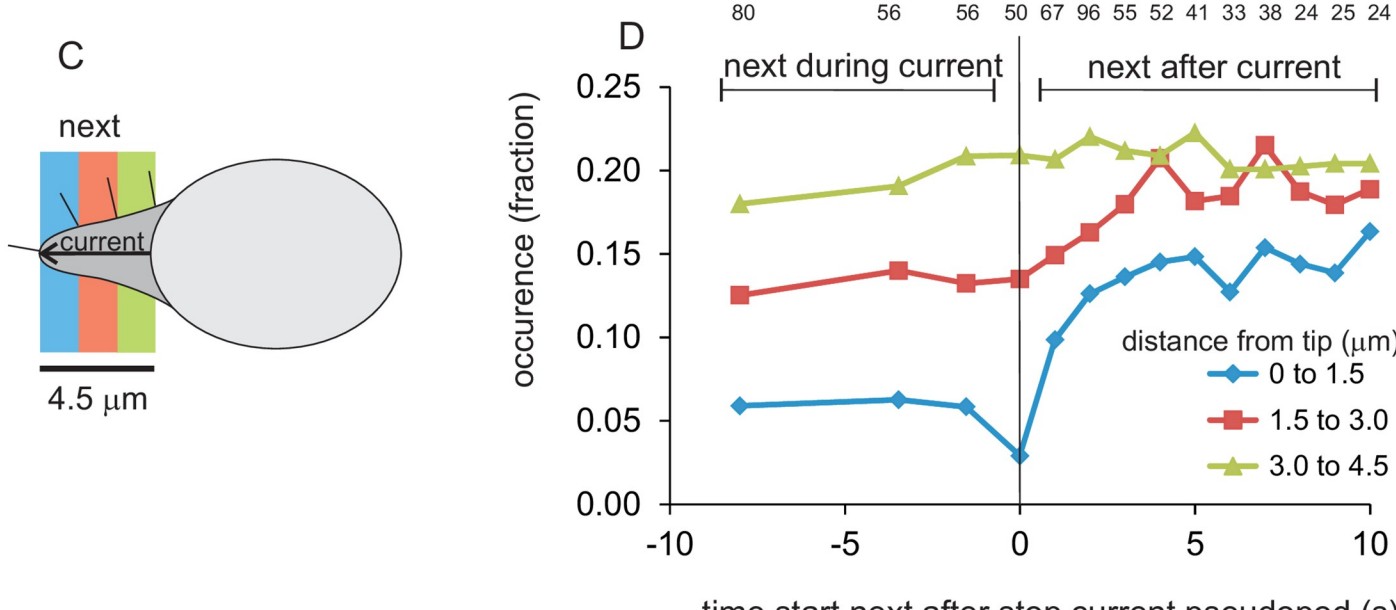

**Fig 3. Pseudopods STOP by local inhibition at the tip of that pseudopod.** (A) Growth time and size of pseudopods in polarized wild-type cells that at start have different number of other pseudopods; the data show the means and SEM; n = number of pseudopods as indicated. (B) Growth time and size of pseudopods in all 16 strains at different number of other pseudopods relative to cells without other extending pseudopods; the data show the means and SEM; n = number of strains as indicated. (C) Schematic of a cell with an extending pseudopod, and in colors the regions of interest for the start of the next pseudopod. (D) Kinetics of the next pseudopods that start at different distances from the tip of the previous pseudopod. For each time interval before and after the STOP of the current pseudopod the number of the next starting pseudopods was recorded, and their distance from the tip of the current pseudopod was measured. The data are presented for each time interval as the fraction of pseudopods starting at the indicated distance, with total number of starting pseudopods at that time interval as indicated by the number above the figure. Pseudopods that start between -11s and -5s are binned and shown at -8s, start between -5s and -2s are shown at -3.5s, start at -2s and -1s are shown at -1.5s; other data points are the pseudopods that start at the time shown.

no discrimination was made between new pseudopods that appear during or after the extension of the previous pseudopod. Here the start of new pseudopods was analyzed with both high temporal and spatial resolution. Temporal information was obtained by determining the number of new pseudopods that start just before or after a pseudopod stops. High spatial

resolution data was extracted by determining the number of new pseudopods that start close to the tip of that pseudopod (0 to 1.5 μm), just below the tip (1.5 to 3.0 μm) or further away from the tip (3.0 to 4.5 μm; see Fig 3C for geometry). Cells start 277 new pseudopods while the current pseudopod is still extending. The results of Fig 3D reveal that only 5.1% of these pseudopods starts within 1.5 μm from the tip of the extending pseudopod, while 13.3% start at a distance of 1.5 to 3.0 μm from the tip and 19.3% at a distance of 3.0 to 4.5 μm from the tip of the still extending pseudopod. Many new pseudopods start after the current pseudopod stops. The probability that these new pseudopods start within 1.5 μm from the old tip rapidly increases 3-fold from 5.1% to 15% after the previous pseudopod stops; at a distance of 1.5 to 3.0 μm from the old tip the probability increases 1.4-fold from 13.3% to 19.2%, while the probability does not increase at a distance of 3.0 to 4.5 μm (Fig 3D). In summary, the mechanism that leads to STOP of the extending pseudopod does not inhibit the side of the extending pseudopod, but inhibits only the extending tip over a length of a few micrometers.

## Pseudopod STOP depends on the time, size and rate of extension

The 996 extending pseudopods have different growth times (Fig 4A). Very few pseudopods have stopped within 5 s, 50% of the pseudopods have stopped after about 10 s and nearly all pseudopods have stopped after 30 s. The probability of the pseudopods to stop after a growth time $t$ ($P_{STOP}$ or $\beta(t)$ in fraction/s) is the number of extending pseudopods that stop in a time interval of 1 second divided by the number of pseudopods that have not yet stopped at the beginning of that time interval. Fig 4B reveals that $\beta(t)$ increases approximately linear with the time of extension of the pseudopods, or $\beta(t) \approx \gamma t$. This suggests that during pseudopod extension inhibitory activity builds up that mediates pseudopod STOP. Consistent with this observation, the kinetics at which the pseudopods stop is approximately linear with the $t^2$ (Fig 4C). The slope (0.5$\gamma$) represents the population rate parameter for stopping yielding $\gamma =$ 0.0124 ± 0.0035 $s^{-2}$ (mean and 95% confidence interval of linear regression with n = 21 time points). The intercept with the x-axis is $t_0^2$ = - 3.33 ± 3.70, which is not significantly different from zero (P>0.1), suggesting that immediately after the start of a pseudopod inhibitory activity begins to builds-up, increases proportional to the time of extension, and eventually leads to pseudopod STOP.

The inhibitory processes leading to pseudopod STOP may be time-dependent (such as synthesis of inhibitory molecules), size-dependent (such as accumulating tension in the membrane or increased bending of the membrane at the tip), or rate-dependent (such as drag of the extending pseudopod). Discriminating between these models requires detailed analysis of the data, because the time and size of the extending pseudopod are connected: as the pseudopod extends, the growth-time and size increase, both proportional to the observed constant rate of extension. In data-driven analysis of pseudopod STOP three pieces of evidence suggest a major role of pseudopod size. First, in a logic reaction of $A^*B \rightarrow C$ the relative error in C must be larger than the relative error in A or B. The relative error (%SD; Fig 1C) in the rate (46%), growth time (52%) and size (37%) suggest that the growing size with its smallest relative error may be the cause of pseudopod STOP and growth time is the consequence. A second indication comes from the powerful technique to analyze subsets of data: the large set of 996 pseudopods was divided into three groups of 332 pseudopods each based on their rate of extension: slow, fast and intermediate. The cumulative fraction of stopped pseudopods reveals that fast, intermediate and slow pseudopods stop at similar sizes (S3A Fig in S1 File), but at very different growth times (S3B Fig in S1 File). A third indication comes from Fig 4D showing an inverse relationship between rate and growth time. This inverse relationship implies that rate-*time is nearly constant (note: rate[μm/s]*time[s] = size[μm]). Thus a pseudopod that extends

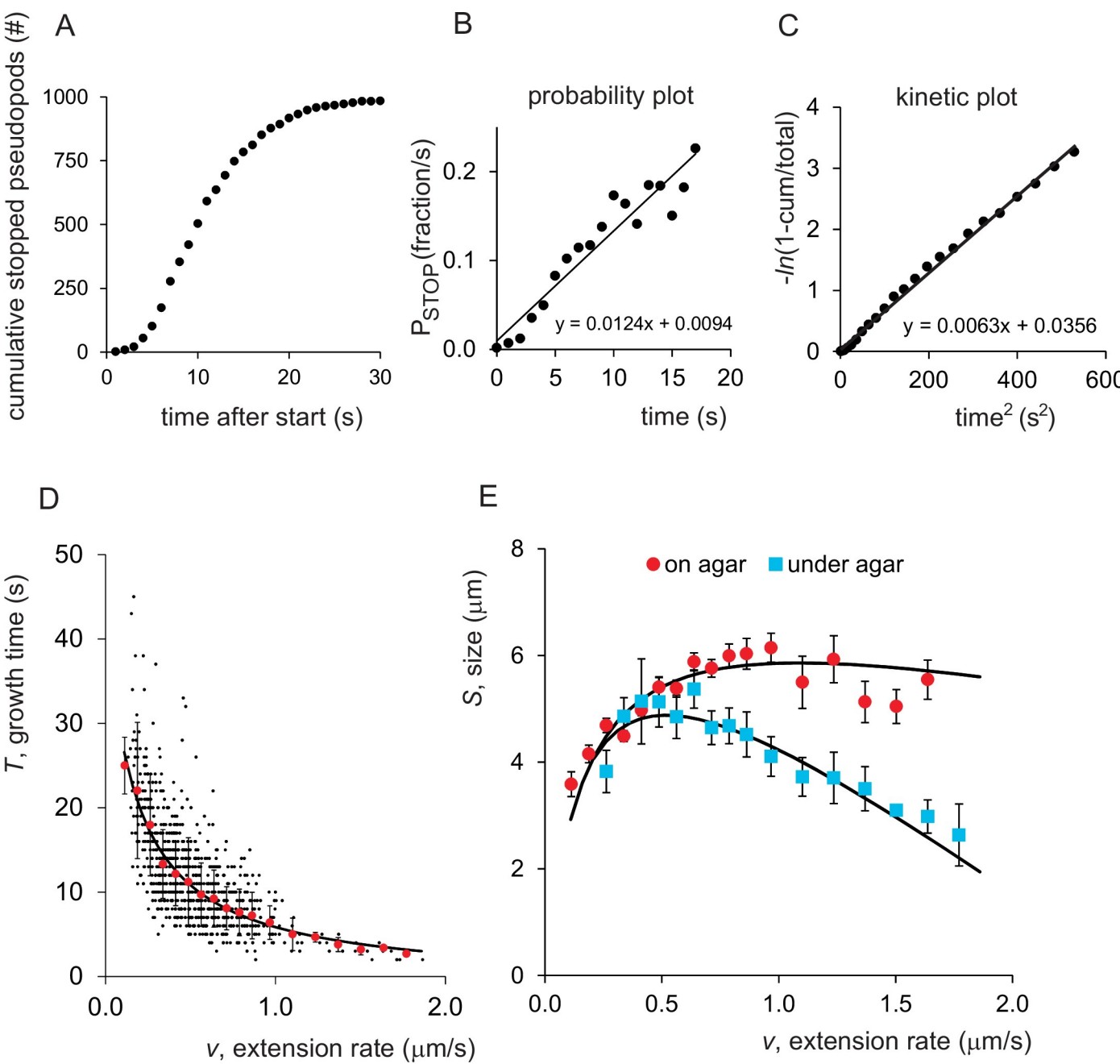

**Fig 4. The STOP of pseudopods is regulated by time, size and rate.** (A) Cumulative number of pseudopods that have stopped at different times after start. (B) The probability that a pseudopod stops ($P_{STOP}$) is defined as the fraction of pseudopods that stop in a 1s time-interval divided by the pseudopods that have not yet stopped at the beginning of that time-interval. $P_{STOP}$ is not constant as $P_{START}$, but increases with the time of extension. (C) Kinetic plots of the same data as a function of time$^2$; cum is the cumulative number of pseudopods that have stopped and total = 996. Linear regression with n = 21 time points yield intercept with the time$^2$-axis of -3.33 ± 3.70 and slope of 0.00629 ± 0.00019 s$^{-2}$ (optimal value and 95% confidence interval of the linear fit). (D) The growth time of a pseudopod is inversely related to the rate of extension; black dots are individual pseudopods, red symbols are means and SD of multiple pseudopods binned for rate intervals of 0.075 µm/s. (E) The size of pseudopods as function of the rate of extension. The data are the means and SEM of multiple pseudopods binned for a specific rate interval. The filled line for wild-type on agar (red symbols) represent the optimal fit of the model according to Eq 1. The blue symbols represent pseudopods of wild-type cells moving under agar; pseudopods stop prematurely, especially at higher rates (see Table 1 for parameter values).

at a low rate stops very late, while a pseudopod extending at a high rate stops very early, but both obtain a similar size (Fig 4D and 4E). These three pieces of evidence strongly suggest that

pseudopod stop not because they extend for a certain time, but predominantly because they reach an inhibitory size. Other aspect such as width of the pseudopod and curvature appear not to have a measurable effect on pseudopod growth time (S4 Fig in S1 File). Although size-dependent inhibition may dominate pseudopod STOP, time-dependent or rate-dependent inhibition may participate under specific conditions. For instance, Fig 4B is not perfectly linear; in addition, S3 Fig in S1 File and Fig 4E reveal that slow pseudopods stop at slightly smaller sizes than intermediate and fast pseudopods. Possibly, in slow extending pseudopods the inhibitory size is reached so late that other inhibitory events may also contribute to pseudopod STOP.

To dissect pseudopod stopping in quantitative detail, the purely data-driven analysis was complemented with a hypothesis-driven analysis using a minimal physical model for pseudopod extension (see Discussion, Fig 6 and S1 Text in S1 File). The model assumes that polymerization of branched F-actin in the extending pseudopod induces a forward activity/force leading to pseudopod extension, which is counteracted by negative activities/forces that increase with size, time and rate of the extending pseudopod. Pseudopods stop when the counterforces become larger than the forward force. This forward force is arbitrarily set at 1, which implies that that pseudopods stop when $k_s s + k_t t + k_v v \geq 1$, where $k_s$, $k_t$, and $k_v$ are the contribution of size $s$, time $t$ and rate $v$, respectively, to the counterforces (see S1 Text in S1 File for theoretical background). Since by definition $t = s/v$, and since $v$ is constant during the extension of a pseudopod (Fig 1B), the size $S$ of a pseudopod at the moment of STOP can be written as

$$S = \frac{1 - k_v v}{k_s + k_t/v} \tag{1}$$

The role of time, size and rate in pseudopod STOP was investigated by fitting the experimental data of the dependence of size on the rate of 996 pseudopods (Fig 4E) to this equation, using models with increasing number of the parameters $k_s$, $k_t$, and $k_v$. The optimal fits were evaluated using statistical methods for model discrimination (see S5 Fig in S1 File). With one parameter, only pseudopod stopping by size-dependent inhibition ($k_s$) can describe the observed data to some extent. However, to explain the smaller pseudopods at low rates, the contribution of time-dependent processes to pseudopod stopping is essential and statistically highly significant (P<0.0001; S5 Fig in S1 File). At rates >1 μm/s the observed sizes are smaller than predicted using a model based on size and time only, but are well explained when a contribution of rate-dependent processes ($k_v$) to pseudopod stopping is incorporated; also the contribution of this third parameter is statistically significant (P<0.001; S5 Fig in S1 File). The potential contribution of a fourth parameter (such as an exponent of the rate $v^z$) was investigated; in all cases the improved fit is too small to be significant (P>0.2) and models with four parameters are rejected (S5 Fig in S1 File). In conclusion, pseudopod stopping is mediated by three components that depend on the increasing size, time and rate of the extending pseudopod. Their relative contribution is given by the three terms in $k_s s + k_t t + k_v v = 1$, indicating that pseudopod stopping is mediated for approximately 68% by size-, 26% by time- and 6% by rate-dependent processes (see S2 Table).

The role of rate in pseudopod stopping could be related to resistance and drag forces experienced by the extending pseudopod in buffer. These forces depend on the rate of pseudopod extension and on the viscosity of the medium. Cells moving under agar may experience more resistance due to the need to deform the poro- and visco-elastic material [26]; more force is needed at a higher speed of deformation (see S1 Text in S1 File). Therefore, pseudopod extension of cells on agar was compared with pseudopod extension of cells under a layer of 1.5%

agar. Under agar, pseudopods are 15% smaller and have a 40% shorter growth time (Table 1). Interestingly, the contribution of time ($k_t$) and size ($k_s$) to pseudopod stopping is unaltered, but the contribution of rate ($k_v$) is strongly increased (Fig 4E). Thus pseudopods that are extended at a low or intermediate rate are hardly affected by the increased resistance of the medium, but fast extending pseudopods are hindered strongly.

In summary pseudopod STOP is a complex inhibitory process that depend on time, size and rate of the extending pseudopod, and these inhibitory processes operate locally at the very tip of the extending pseudopod.

**Steady-state number of extending pseudopods.** The 996 extending pseudopods were recorded over a total period of 15,356 seconds. During about 28% of the time cells have no extending pseudopods ($C_0$), the majority of the cells have one extending pseudopod ($C_1$), while cells with the maximum of four extending pseudopods were observed during only 6 seconds. This observed steady state distribution for the number of extending pseudopods per cell was compared with the expected distribution predicted from the rate constants $\alpha_n$ of formation of the $n^{th}$ pseudopod and the rate constants $\beta_n$ of stopping of the $n^{th}$ pseudopod (Fig 5A). The kinetic experiments on pseudopod START have shown that $\alpha_n$ decreases by a factor A for each pseudopod present ($\alpha_n = \alpha_1 / A^{n-1}$), while the kinetic experiments on pseudopod STOP reveal $\beta_n$ has the same value for all pseudopods ($\beta$). On a microscopic scale of a few seconds pseudopod STOP $\beta$ is a complex function of size, time and rate; however on a macroscopic scale of several hours STOP can be described as a population rate parameter, $\tilde{\beta}$, which is related to the kinetics of stopping and is given by $\tilde{\beta} = \sqrt{0.5\gamma}$ (Fig 4C and S1 Text in S1 File). Therefore, the steady state distribution for the number of extending pseudopods per cell depends on $\alpha_1 / \tilde{\beta}$ and A.

When all pseudopods have the same stochastic START (A = 1), the expected number of extending pseudopods is a wide distribution, including many cells with multiple pseudopods (Fig 5B). At increasing values of A the fraction of cells with multiple pseudopods strongly decreases and the majority of cells will have only one extending pseudopod. The observed distribution for polarized cells is best described by $\alpha_1 / \tilde{\beta} = 1.79 \pm 0.13$ and $A = 3.55 \pm 0.32$ (means and 95% confidence interval; Fig 5C, S2 Table in S1 File). This value of A for the steady-state distributions of pseudopods is very close to the value obtained from kinetic experiments (Fig 2), which yielded $A = 3.59 \pm 0.28$. Fig 5C also presents the pseudopod distribution of two mutant cells, to be discussed below, that are either enriched in one pseudopod (*pla2*-null with A = 12.8) or have many pseudopod (*forAEH*-null with A = 1.4).

## Cell polarization and chemo-attractants do not affect pseudopod dynamics

The extensive analysis of pseudopod dynamics described above was performed on 6 hours starved *Dictyostelium* cells that are polarized with a relatively stable active front and a relatively inactive side and rear of the cell [6,7]. Pseudopod dynamics was also measured for unpolarized wild-type cells that were starved for 3 hours. These cells do not have a stable front and extend pseudopods from all regions of the cell. Unpolarized cells move in nearly random directions, in contrast to polarized cells that move with strong persistence of direction [6,7]. The primary data obtained for unpolarized cells are growth time, rate, size and pseudopod interval; the deduced data are the basal probability to start a new pseudopod ($\alpha_1$), the inhibition of the start of a new pseudopod during the extension of the current pseudopod (A), the population rate parameter to stop pseudopod extension ($\tilde{\beta}$), and the time-, size- and rate-dependent kinetic constant to stop ($k_t$, $k_s$ and $k_v$). Surprisingly, pseudopod dynamics is not statistically significantly different between unpolarized and polarized cells for nearly all parameters mentioned

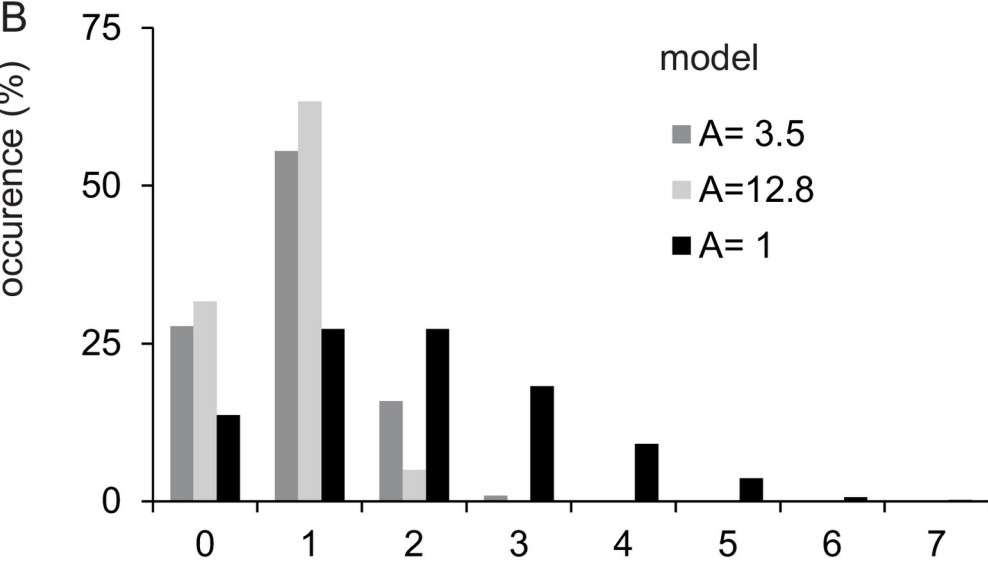

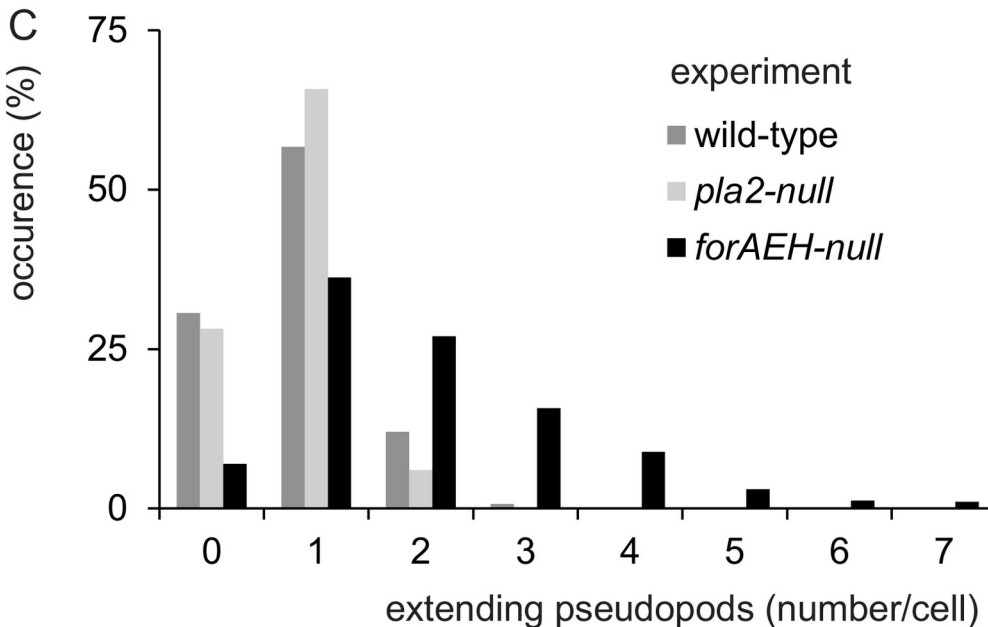

**Fig 5. Number of extending pseudopods.** (A) Schematic of pseudopod extension as deduced from the analysis of START and STOP. $C_n$ is a cell with $n$ extending pseudopods. The formation of a pseudopod is given by the rate constant $\alpha$, that is reduced $A$-fold by each extending pseudopod (see Fig 2). The termination of pseudopod extension is given by the macroscopic rate constant $\tilde{\beta}$ that is independent of the number of extending pseudopods (Fig 3A and 3B);

a cell with e.g. three pseudopods has three possibilities to stop one of its pseudopods, giving $3\tilde{\beta}$. (B) Prediction of the number of extending pseudopods using $\alpha/\tilde{\beta} = 2.0$, and different values of $A$. (C) Experimental observations for polarized wild-type and two mutants. Equations S17 and S18 in S1 Text in S1 File were used to predict (panel B) or to fit experimental data (panel C); the optimal value and 95% confidence interval for the fitted values of $\alpha/\tilde{\beta}$ and $A$ are given in S2 Table.

above (see Table 1). The only statistically significant difference is the absence of pseudopod STOP by the rate component ($k_v$).

Polarized cells can move in the direction of the chemoattractant cAMP. Cells were exposed to a shallow gradient of cAMP (0.5 nM/μm) that induces a chemotaxis index of 0.62 [13]. Interestingly, none of the kinetic parameters determined is statistically significant different between polarized cells moving in random directions in buffer and cells moving directionally in a gradient of chemoattractant (Table 1). In conclusion, cell polarization and chemoattractants influence the position at the cell where pseudopods are made: in the front of polarized cells where previous pseudopods were made, or at the side of the cell exposed to the highest concentration of chemoattractant. Importantly, in both cases the timing of START and STOP of pseudopod extension is identical and not affected by internal or external spatial cues of polarized and chemotaxing cells, respectively.

## Pseudopod dynamics of mutants begin to reveal molecular mechanisms

Pseudopod formation has been measured for many signaling mutants either in buffer, in a chemotactic gradients or in an electric field [5,7,9,13,18,27–32]. Many proteins have been identified that regulate the position of pseudopod extension, but only a few mutants have altered pseudopod dynamics, which are described in detail here (see Table 1).

The Scar/Wave complex consists of five proteins, Scar, PIR121, Nap1, HSPC300, and Abi that functions as a signaling hub; Rac is one of the upstream activators. Activated Scar complex recruits and activates the Arp2/3 complex to induce branching of F-actin [33–36]. Cell size and speed, pseudopod size, and actin polymerization are all decreased in mutants lacking the Scar protein [33,36,37]. Current analysis reveals that *scar*-null cells extend very small pseudopods with very short growth times, but at a nearly normal extension rate. The START of pseudopods ($\alpha_1$) is not impaired by deletion of Scar, but actually strongly increased 2.5-fold (Table 1). The extending pseudopods inhibit the START of new pseudopods as in wild-type cells ($A$). The main defect of *scar*-null cells is a very strong 4.5-fold increase of pseudopod STOP ($\tilde{\beta}$), which is caused by a strong increase of stopping by size ($k_s$) and by time ($k_t$). Since both START and STOP are increased, the average number of extending pseudopods is only slightly altered in *scar*-null cells compared to wild-type cells, explaining why *scar*-null cells move relatively well, although at reduced speed [33,36,37].

PLA2 was identified in screens for proteins involved in chemotaxis [38–40]; *pla2*-null cells in the absence of chemoattractant appears to have larger pseudopods, which is associated with a longer growth time at a normal extension rate ([7,13] and Table 1). The START of pseudopods is normal ($\alpha_1$), but with a strongly increased 12.8-fold inhibition of the start of new pseudopods by extending pseudopods ($A$). Consequently, *pla2*-null cells very rarely have more than one extending pseudopod (Fig 5C). The increased pseudopod size is caused by the reduced STOP, which is due to a strong decrease of stopping by size ($k_s$) and by time ($k_t$). In many respects, pseudopod kinetics of *pla2*-null cells exhibit the opposite phenotype compared to those of *scar*-null cells.

Myosin filament formation in the rear of *Dictyostelium* cells is regulated by a cGMP-signaling pathway composed two guanylyl cyclases GCA and sGC, the cGMP-binding protein GbpC

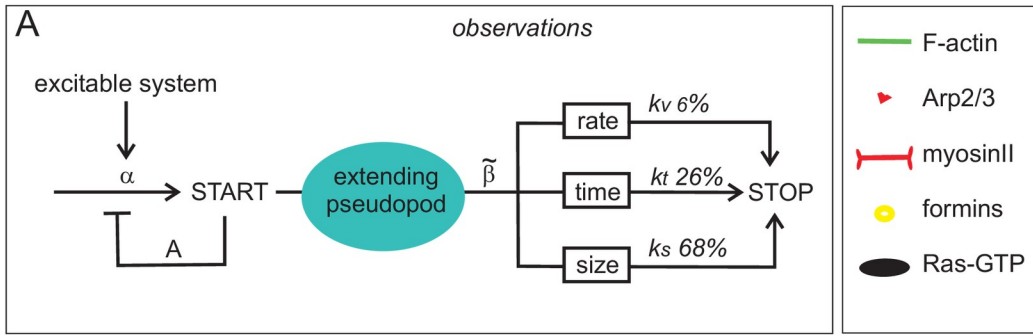

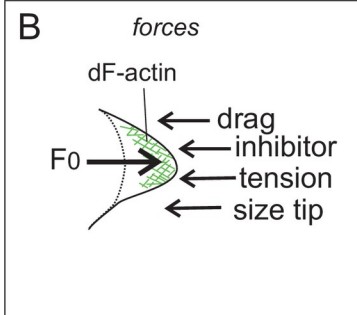

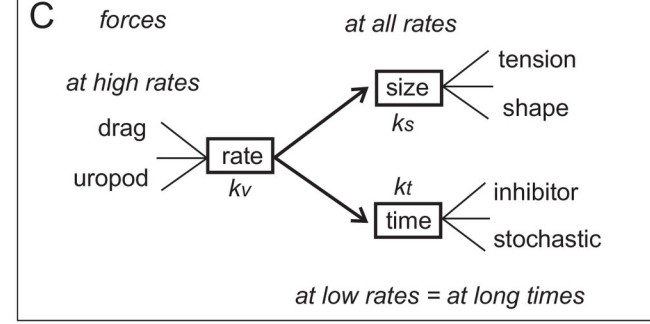

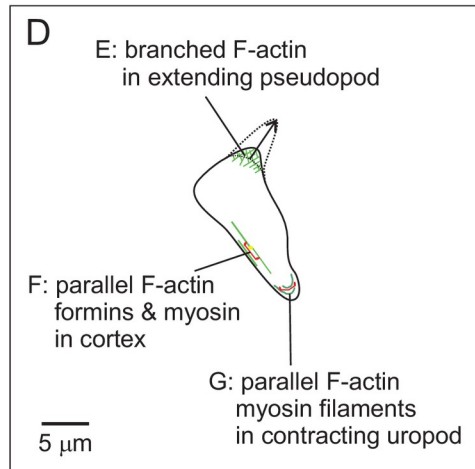

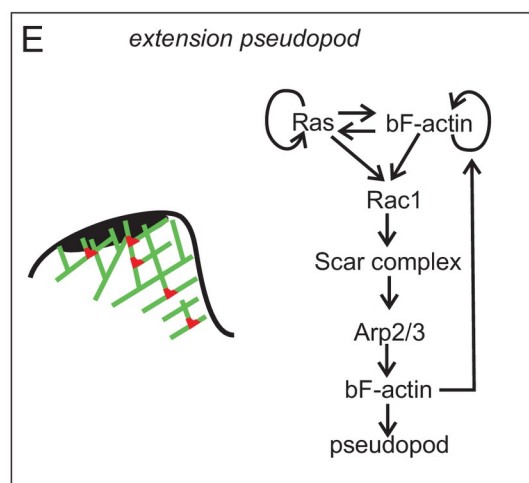

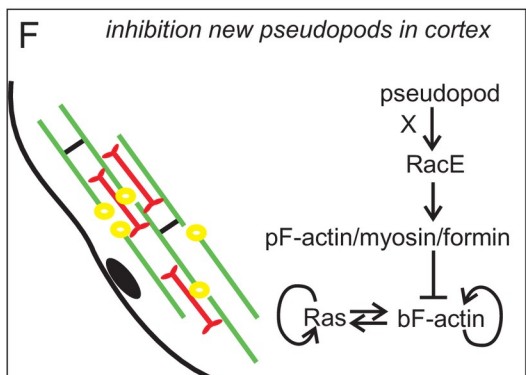

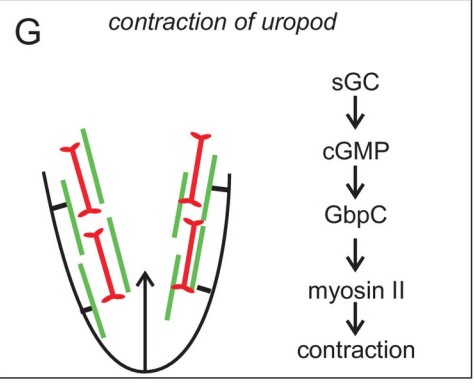

**Fig 6. Summary and unified model of pseudopod extension.** (A) Schematic of pseudopod extension for all stains and conditions. An excitable system triggers the START of pseudopod extension with rate constant $\alpha$. Each extending pseudopod inhibits the start of a new pseudopod $A$-fold. The pseudopod extends at a rate $v$, thereby -with time $t$- reaching

a larger size $s$. The STOP of the pseudopod is mediated by processes that depends on a combination of rate, time and size; together they represent a macroscopic rate constant of stopping $\tilde{\beta}$. The relative contribution of rate, time and size to STOP is given for polarized *Dictyostelium* cells. Panels B and C present a physical model. (B) The strong polymerization of branched F-actin (bF-actin) induces a forward force; at the start of pseudopod extension, the force $F_0$ is mediated by about 4000 bF-actin filament at the emerging tip (see S6 Fig in S1 File). (C) The experiments reveal that multiple counterforces contribute to pseudopod STOP; see text for details. Panels D-G presents the outcome of mutant analysis in *Dictyostelium*. (E) The extending pseudopod. In *Dictyostelium* a Ras-bF-actin-excitable system triggers pseudopod extension by activating Rac1 and the Scar complex, which induces Arp2/3-mediated actin nucleation and branching; *scar*-null cells stop prematurely. (F) Inhibition of new pseudopods. The cortex is a ~100 nm thick sheet under the plasma membrane consisting of parallel F-actin filaments (pF-actin), myosin filament and additional F-actin-binding proteins; formins stimulate and stabilize the pF-actin/myosin structure. Branched-F-actin is not easily formed in a strong cortex. Observations reveal that an extending pseudopod very fast generates an unknown global signal X that inhibits new pseudopods. This inhibition requires RacE and the pF-actin/formin/myosin cortex (cluster **a** in S7 Fig in S1 File). (G) The uropod. In *Dictyostelium*, a cGMP-based signaling pathway activates the interaction between myosin filaments and pF-actin filaments, leading to contraction of the uropod. Pseudopods that extend at a very high rate stop prematurely in cells with a very strong uropod (cluster **b** in S7 Fig in S1 File). In mammalian cells the three major components of the cell -pseudopod, cortex and uropod- play similar roles as in *Dictyostelium*, but the excitable system in panel E may have other small GTP-proteins such as CDC42, and the uropod in panel G is not regulated by cGMP but by the Rho-kinase Rock.

and myosin II [22]. Mutants defective in these proteins have reduced polarity, make more pseudopods at the side of the cell and lack a strong back of the cells with a retracting uropod [7,41]. In mammalian cells, these functions are attributed by RhoA and the Rho-kinase ROCK [42,43]. Mutants lacking the two guanylyl cyclases (*gc*-null), or the only cGMP-target GbpC have similar pseudopod properties: an increase of START ($\alpha_1$) and the absence of STOP by rate ($k_v$). Myosin II null cells have similar properties for $\alpha_1$ and $k_v$, but additionally show a defect in the inhibition of new pseudopods by extending pseudopods ($A$). The defects of these mutants are statistically highly significant but relatively mild and therefore size, growth time and number of extending pseudopods per cell are not very different from wild-type cells.

Inhibition of myosin filament formation in the front of *Dictyostelium* cells is regulated by Rap1GTP that is activated in the front half of the cell [44,45]. Wild-type cells expressing the dominant active form Rap1G12V exhibit strongly reduced myosin filaments and cortical F-actin and extend multiple pseudopods [44]. Pseudopod analysis (Table 1) reveals that Rap1G12V cells extend multiple small pseudopods, due to an increase of START ($\alpha_1$), weak inhibition of the START of new pseudopods by an extending pseudopod ($A$), and by increased STOP by rate ($k_v$) and size ($k_s$). This phenotype mimics that of myosin II null cells, but are much stronger. In contrast to *myoII*-null cells, Rap1G12V cells exhibit increased adhesion to the substratum, which may add to the pseudopod phenotype [46].

Recently it was reported that mutants with a deletion of three formins (*forAEH*-null) or a deletion of RacE have pronounced pseudopod activity [47]. These proteins regulate the cortex of parallel F-actin/myosin at the side of the cell [47,48]. The mutants *forAEH*-null and *racE*-null have very similar pseudopod dynamics (Table 1). These cells START pseudopods at about 50% elevated rate ($\alpha_1$). Cells have shorter pseudopods due to an increase of STOP by rate ($k_v$) and size ($k_s$). Importantly, cells are strongly characterized by the very low inhibition of the START of new pseudopods by an extending pseudopod ($A$). Since cells START at slightly elevated rate ($\alpha_1$), the absence of inhibition of new pseudopods lead to the simultaneous extension of many pseudopods; cells contain up to 7 extending pseudopods (Fig 5C). As a consequence, cells movement appears chaotic with multiple extending pseudopods in different directions [5,47]. The parallel F-actin cortex at the side of the cell also contains myosin II filaments [49]; *myoII*-null cells share a reduced inhibition of new pseudopods ($A$) with *forAEH*- and *racE*-null mutants. Interestingly, the pseudopod phenotype of *forAEH*- and *racE*-null mutants is very similar to cells expressing dominant Rap1G12V (Table 1), suggesting that Rap1-GTP may inhibit the entire contractile cortex in the front of the cell including myosin filaments.

The leucine-rich-repeat protein LrrA is a scaffold connecting heterotrimeric and monomeric G-proteins in *Dictyostelium* [50,51]; the mammalian protein Shoc may have a similar function [52,53]. The scaffold LrrA coordinates in time and space the activation of multiple G-proteins (G$\alpha$, G$\beta\gamma$, Ras, Rap, Rac) of cells in buffer and in chemotactic gradients, and thereby regulates many down-stream signaling pathways. *lrrA*-null cells in buffer have reduced Ras and increased Rap and Rac activation, and exhibit enhanced pseudopod activity [51]. Detailed pseudopod analysis confirms the complex phenotype of these cells, as nearly all aspects of pseudopod dynamics are altered in *lrrA*-null cells (Table 1): smaller size and shorter growth time due to enhanced STOP ($\tilde{\beta}$), enhanced START ($\alpha_1$), and especially reduced inhibition of new pseudopods by extending pseudopods (*A*). Although nearly all pseudopod parameters are altered, the fact that both START and STOP are enhanced results in only a mild increase of the steady-state number of extending pseudopods; *lrrA*-null cells can move and displace relatively well, in contrast to the *forAEH-* and *racE*-null mutants with multiple extending pseudopods.

## Pseudopod extension in other cell types and species

The method of dissecting pseudopod dynamics was used to explore how cells from other organisms move. Three cell lines were analyzed: mammalian neutrophils that move at a high rate and exhibit chemotaxis as *Dictyostelium* cells [2,54,55], mammalian mesenchymal stem cells that make spiky protrusions and move at a very low rate [56], and the fungus *B.d. chytrid* that has a Scar-regulated polymerization of branched F-actin in protrusions [20].

Pseudopod dynamics in neutrophils is very similar to that of *Dictyostelium* cells, with subtle differences in size, growth time and rate (Table 1). The START of pseudopods is a stochastic process with strong inhibition of the start of a new pseudopod by an extending pseudopod. The STOP is regulated by a combination of size, time and rate, similar as in *Dictyostelium*. The distribution of the number of extending pseudopod is also similar. We conclude that pseudopod dynamics in *Dictyostelium* and neutrophils are essentially identical, except for the numerical differences that are relatively small considering the different optimal temperatures (22 versus 37 °C) and long evolutionary distance between these organisms (2 billion years).

Mesenchymal stem cells have a few properties different from the other cell lines. These cells move extremely slow and pseudopods extend at a very low rate of 0.05 μm/s and have a very long growth time of 460s. Although the beginning and end of pseudopod extension was easily detectable, during pseudopod growth the speed of the tip showed more variation that in other cell lines with repeated of faster and slower pseudopod extension. The mean rate of pseudopod extension is 16-fold lower that the rate in neutrophils, the growth time is 90 fold longer and pseudopods are 4-fold larger than in neutrophils (Table 1). In addition, the shape of the protrusions is very spiky. After the period of extension, the protrusions of mesenchymal cells adhere to the substrate, are not retracted or rapidly filled with cell body, and therefore in still images these cells have multiple protrusions (3.4 per cell, Table 1), giving their dendritic appearance. However, on average a mesenchymal cell has only 0.38 extending protrusions (Table 1) and most protrusions on still images are stationary. Despite these strong differences with other cells, START is similar with first order kinetics and strong inhibition by extending protrusion (*A*). STOP appears to be exclusively regulated by the size of the extending protrusion; the obtained fitted values for time ($k_t$) and rate ($k_v$) are statistically not significantly different form zero, indicating that these very slowly extending protrusions stop by size-dependent inhibition only.

*Batrachochytrium dendrobatidis chytrid* (*B.d. chytrid)* is a fungus that infects the skin of Amphibians. Although cells have a flagellum, they mainly move with protrusions. Fritz-Laylin et al [10] have shown that *B.d. chytrid* is an example of organisms in which the presence of

genes encoding for Scar and/or WASP is associated with movement by pseudopods. Pseudopod analysis reveals that the rate of pseudopod extension in *B.d. chytrid* is similar to that of *Dictyostelium* and neutrophil cells. The pseudopod growth time (5 s) is relatively short and therefore pseudopods are small (2.2 μm). The START rate ($\alpha_1$) is high compared to other organisms, but as in other organisms, kinetics is first order and strongly inhibited by extending pseudopods (*A*). Pseudopod STOP rate ($\tilde{\beta}$) is also relatively high, and therefore pseudopod extension is very dynamics, but cells still extend predominantly only one pseudopod at a time. Pseudopod STOP is mediated by a combination of mainly size and partly time and rate, as in *Dictyostelium* and neutrophils.

## Discussion

Previous work in *Dictyostelium* and other organisms have shown that pseudopods are extended by polymerization of branched F-actin [57–66]. Branched actin networks require the Arp2/3 complex that is activated downstream of the SCAR/WAVE complex and the Rac family of GTPases [1,57,58,63,67–69]. The localized polymerization of branched actin occurs in an excitable medium of bF-actin and small GTPases [29,65,69–76], which means that small fluctuations of the excitable medium are damped, but larger fluctuations above a threshold are amplified leading to a large area of pseudopod-inducing activity. Delayed inhibition induced by the extending pseudopod leads to termination of pseudopod extension. In the present study information on the kinetics of pseudopod extension was collected for a very large data set that therefore includes sufficient information on rare events to allow dissection of causal relationships and to provide mechanistic insight how pseudopods start and stop. The basis of this mechanistic understanding is the notion that pseudopod START and STOP is regulated by an ON/OFF switch. Current and previous observations [7] suggest that the ON switch is a stochastic event in which the excitable medium of bF-actin/Ras [29,65,69–76] surpasses a threshold. We and others [29,45,71,74,76,77] have proposed that the OFF switch is due to the opposite reaction: the extending pseudopod produces local and/or global inhibitors by which the activity of the excited medium declines below the threshold level for excitation, and consequently pseudopod extension STOPs. However, the current observation indicate that STOP is not simply the opposite of START, but is regulated by different and independent mechanisms. The experiments suggest that the extending pseudopod produces at least two inhibitors, which can be inhibitory molecules, depletion of substrate, or a physical property that counteracts extension. First, a very global inhibitor that suppresses the START of a new pseudopod in the entire cell; the strength of this global inhibitor increases with the power of the number of extending pseudopods, and presumably reduces the excitability to induce new pseudopods. Surprisingly, this global inhibitor of START has no effect on the STOP of extending pseudopods: the observed growth time and size of extending pseudopods is independent of the number of other extending pseudopods, even in mutants that extend five pseudopods simultaneously. The second inhibitor produced by the extending pseudopod operates very local and mediates the STOP of that pseudopod; its strength depends mainly on the increasing size of the extending pseudopod. Importantly, these properties of inhibition of START by global inhibition and STOP by local pseudopod size is observed in four organisms/cell lines: fast moving chemotactic protist *Dictyostelium* and mammalian neutrophils, the slow moving mammalian mesenchymal stem cells and the flagellum/pseudopod containing fungal *B.d. chytrid*. These apparently fundamental properties of the regulation of START and STOP is instrumental for these amoeboid cells to extend predominantly only one pseudopod at the time and that all pseudopods have a similar size, which greatly increases the efficiency of cell movement.

## START

In *Dictyostelium* new pseudopods are formed in an coupled excitable medium of bF-actin and Ras [29,45,78]. In mammalian cells excitable Cdc42 may have a similar role as Ras [79]. The start of pseudopod extension is preceded by a local increase of F-actin and Ras-GTP in *Dictyostelium* [45], and Cdc42 in neutrophils [79]. The present data reveal that naïve *Dictyostelium* cells that have no pseudopods will START a first pseudopod randomly with a probability of 14% per second. The START of a second pseudopod is inhibited about 3.5-fold. The appearance and disappearance of this inhibition was carefully analyzed. If START of a second pseudopod is not inhibited and occurs also with a probability of 14%/s, the expectation is that when 622 naïve cells without pseudopods start a first pseudopod, there is a 14% change that they start a second pseudopod in the same frame (i.e. 87 cells). However, we observe that only 25 naïve cells start two pseudopods in the same frame, which is 3.5 fold less (Fig 2C). Therefore, the 3.5 fold inhibition of a second pseudopod is already present before the first pseudopod has extended. After the first pseudopod stops, the probability to START a new first pseudopod recovers immediately to 14%/s (Fig 2C). Even in the special cases where two pseudopods STOP simultaneously, the very low 1% probability to start a third pseudopod has recovered immediately to 14%/s when the two pseudopods stop (S2 Fig in S1 File). What can be the nature of such a global inhibitor that appears and disappears within one second after start or stop of a pseudopod, respectively? This inhibition has similar values in neutrophils, mesenchymal cells, *B.d. chytrid*, and *Dictyostelium* wild-type and several mutants. The inhibition of the start of a new pseudopod by an extending pseudopod has also similar values in cells moving against increased resistance under agar ($A = 3.69 \pm 0.42$; Table 1) in a gradient of chemoattractant ($A = 4.06 \pm 0.88$) or in an electric field ($A = 3.32 \pm 0.29$; movies 5 and 6 from reference [27]). However, it is strongly reduced in *myoII*, *forAEH* and *racE*-null mutants, and dominant active Rap1G12V cells, all with a disruption of the cortical parallel pF-actin/myosin cytoskeleton (cluster **a** in S7B Fig in S1 File; Fig 6F). The formins stimulate polymerization of pF-actin [47,58], while myosin II filaments together with pF-actin cross-linkers and pF-actin membrane anchors form a rigid and relatively stable about 100 nm thick cortical sheet under the plasma membrane [58,80,81]. Importantly, it has been demonstrated that this cortical sheet of pF-actin/myosin inhibits the Scar/Rac-mediated formation of branched F-actin to induce a new pseudopod [82–84]. Previously we observed that the excitable Ras-bF-actin is activated locally about 2–3 seconds before the start of the pseudopod [29]. So it is possible that activated Ras stimulates bF-actin in the emerging pseudopod and nearly simultaneously activates pF-actin in the entire cortex to inhibit the formation of a new pseudopod. This inhibitor has to be identified; it should appear extremely fast and act globally. It has been observed that the extending pseudopod enhances global membrane tension [85–87] and cortex tension [88–91], while force on the cortex leads to enhanced formin-induced actin polymerization [92–95], thereby stabilizing the cortex and inhibiting new pseudopods. Although tension is a very attractive candidate for the inhibitor, it is unclear whether tension is exclusively the consequence of pseudopod extension (and then can contribute only after the pseudopod grows) or is also be induced before that pseudopod grows. Recently it was observed that negatively charged lipids are enriched at the inner leaflet of the plasma membrane in the back of the cell [96], where it may stabilize the cortex. It will be interesting to investigate changes in negative charge distribution during pseudopod formation. Occasionally, cells start a pseudopod in the inhibiting cortex. The mechanism how they start may uncover the mechanism of inhibition. A new pseudopod starting in the cortex is always preceded by first a strong local activation of Ras-GTP and later by bF-actin [29]. A plausible model for the start of a pseudopod at the side of the cell is that a local patch of active Ras-GTP activates Rap1 [97] which via Phg2 locally

induces the depolymerization of myosin II filaments [44,98,99] and weakening the pF-actin cortex [49], as suggested by the pseudopod characteristics of cells expressing dominant active Rap1G12V. Therefore, the potential inhibitor may prevent START of a new pseudopod by several mechanisms i) it may reduce the excitability of Ras/bF-actin so that bF-actin is not formed, ii) it may prevent the Ras/Rap- mediated myosin depolymerization so that the cortex is not weakened and bF-actin cannot induce a pseudopod, iii) it may directly stabilize the cortex by e.g. formin-induced pF-actin polymerization, or iv) it may impair the forward force of branched F-actin so that the starting protrusion cannot grow to a pseudopod.

**A model for pseudopod extension.** A conceptual framework of pseudopod extension is used here to explain the observations. Pseudopods are induced by the nucleation and growth of actin filaments in a branched network. The Scar-activated Arp2/3 complex induces new branches of actin filament (Fig 6E). The density of branched F-actin (bF-actin), determined by cryo-electron tomography, is about 300 F-actin filaments per $\mu m^2$ pointing to the membrane [59,60,66] (see S6 Fig in S1 File for geometric details). Initially the surface area of the emerging pseudopod is about 14 $\mu m^2$ containing in the order of 4000 actin filaments pointing towards the membrane. To allow further actin polymerization at the end of these actin filaments near the membrane, an "open" end between an actin filament and the membrane should be available of sufficient space and for sufficient time [100,101]. This can be achieved by three methods: i) stochastic/thermal movement of the membrane and polymerization in the periods of sufficient open space (racket model [102–104]), ii) stochastic/thermal bending of the actin filaments that allows open space [105], and iii) a combination of these two with cooperativity between adjacent actin filaments near the membrane. In each of these models actin polymerization produces a forward force leading to the growth of the pseudopod, while pseudopod growth induces counterforces, such as membrane tension and viscous drag, and counteractivities, including depletion of materials [101]. For the pseudopod to be extended, the forward force should be larger than the counterforces/activities. The counterforces/activities are divided over the number of growing actin filaments in the tip of the protrusion. As long as new actin filaments branch off by new Arp2/3 complexes, sufficient actin filaments remain polymerizing to overcome the experienced counterforces/activities. When the polymerization of some actin filaments is stalled, the counterforces/activities are divided over less filaments, thereby increasing the probability that more filaments fail further extension: the polymerization collapses and pseudopod extension rapidly stops. This model postulates a sharp transition between constant actin polymerization and its sudden arrest, which explains the observation that pseudopods either move at a constant rate, or stop. The model also postulates three major effectors of pseudopod growth and termination, i) Scar-Arp2/3 activity and other factors that enhances actin polymerization, ii) the number of actin filaments in the tip pointing to the membrane, which is proportional to the surface area at the tip of the extending pseudopod, and iii) the emerging counterforces which reduce the probability of an "open" space for sufficient time and size to allow further polymerization of F-actin.

This model explains many observations on the dynamics of pseudopod formation. First, tomography has revealed that new branches are formed under the membrane, and that the branched bF-actin filaments are degraded a few $\mu m$ below the tip of the pseudopod [59,60,66,106], which predicts that the balance of forces that discriminate between either continuation or termination of pseudopod extension operates at the very tip of the extending pseudopod only. This explains the surprising observation that the growth time of a pseudopod is insensitive to the presence of other pseudopods. Second, the forward force of branched F-actin polymerization and pseudopod extension depends on Scar-Arp2/3 activity, which explains the extreme small pseudopods in *scar*-null cells. Third, any chemical or physical process that affects the balance of forward force and counterforces will have an influence on

pseudopod STOP, which can explain why pseudopod termination is regulated by a combination of size-, time- and rate-dependent processes. What is the nature of these inhibitory processes? Below two potential explanations each are given for the role of size, time and rate, respectively, but probably other chemical and physical processes are also relevant (see Fig 6C). The dependence on size may be due to increased membrane tension in the extending pseudopod, similar to the tension in a spring that is proportional to the length of spring extension; tension in the membrane at the tip will reduce the thermal movement of the membrane that is needed to create space for F-actin polymerization. The dependence of STOP on size may also have a geometric cause, because as the pseudopod extends, the surface area of the tip becomes smaller and less actin filaments are present that have to overcome the experienced counterforce (see S6 Fig in S1 File for details): Initially the surface area of the emerging pseudopod contains in the order of 4000 bF-actin filaments pointing towards the membrane, which reduces to about 800 bF-actin filaments in the extended pseudopod. The dependence of pseudopod STOP on time may be due to the production of an inhibitor that accumulates with time in the extending pseudopods; time-dependent inhibitors are an essential component of nearly all pseudopod models [29,71,107,108]. The dependence on time may also have a pure stochastic cause, because the growth of the bF-actin network at the membrane may spontaneously stop at a very low probability, which becomes significant and detectable in pseudopods that extend at a very low rate for a very long time period. The dependence of pseudopod STOP on the rate of pseudopod extension may be due to drag forces that are induced in an extending pseudopod; drag force at low Reynold numbers is proportional to the rate of movement of the small object [109,110]. We noticed that STOP by rate is not detectable in mutants or conditions where cells have no uropod (cluster **b** in S7C Fig in S1 File). The uropod is the stable stiff back of the cell, composed of parallel F-actin and myosin filaments, that generates a force to move the rear of the cell forward. At a high rate of forward movement of the pseudopod, the slower forward movement of the stiff uropod may induce strain in the pseudopod; cells without a stiff uropod may have a more flexible rear that can catch-up with a fast extending pseudopod. As mentioned above, it is entirely conceivable that additional chemical and physical factors are involved in termination of pseudopod extension.

The STOP of pseudopod extension is essentially identical in *Dictyostelium*, neutrophils and *B.d. chytrid* with similar relative contributions of size, time and rate. Mesenchymal cells exhibit different behavior that may be related to the extreme slow extension of the pseudopod. Here pseudopod STOP exclusively depends on pseudopod size. The growth time, which is extremely long with 460 seconds, does not play a role. Perhaps the stability of the F-actin network is strongly increased in mesenchymal cells to allow for these extreme long extension periods. The rate of extension also does not play a role; the rate is extremely slow at 0.05 μm/s, and is unlikely to generate a detectable drag force.

The methods developed here for kinetic analysis of protrusion may also be useful to investigate other actin-based cell deformations, including other outward movements such as filopods or inward deformations; especially interesting are inward deformations which have a defined beginning and end, and have of a limited number per cell such as endosomes or macropinosomes. Is the start stochastic? Is the life of a macropinosome self-regulated such as the stop of a pseudopod. Is the start of a new macropinosome inhibited by active macropinosomes? Is the start of a pseudopod inhibited by an active macropinosome and *vice versa*? In this respect it is interesting to note that in vegetative *Dictyostelium* cells, which have active micropinocytosis, pseudopod formation is enhanced by inhibiting macropinosomes [111].

**Implications.** *Dictyostelium* cells and neutrophils have similar functions: amoeboid movement with high persistence combined with exquisite sensitivity to move directionally in shallow spatial gradients of chemoattractant. Nearly all properties of the pseudopod cycle

uncovered in *Dictyostelium* are also recognized in neutrophils. Efficiency of persistent cell movement is strongly enhanced when cells move with only one pseudopod at the time, which appears to be mediated by the strong inhibition of START by an extending pseudopod. Cells moving directionally in a gradient of chemoattractant exhibit pseudopod dynamics that are identical in all respects to cells moving in buffer. Therefore, pseudopod dynamics is an intrinsic property of the cell: the cells start and stop pseudopods independent of extracellular signals: Cells make pseudopods in a specific rhythm. Extracellular signals that steer cell locomotion do not change this rhythm, but only bias the place where a pseudopod starts, i.e. spatial signals combine with the endogenous excitable pseudopod-inducing activity to bias directional pseudopod extension. Such a system can be extremely sensitive, because in principle each signal-induced activating molecule can provide a small spatial bias to the anyhow emerging pseudopod. In addition, the same system can be used for very different mechanisms of directional movement, including single cell chemotaxis for chemical signals [24], electrotaxis for electric signals [32], haptotaxis for contact signals [112,113], and collective cell migration [114]. Further analysis in other mutants or conditions may uncover additional molecular mechanisms, while pseudopod analysis of other cell types may uncover fundamental aspects of the transition of cell types, including maturation of stem cells in slow moving macrophages or fast moving neutrophils, the mesenchymal-amoeboid transition of cancer cells [115,116], or the induced changes of cancer cell movement by cancer-associated fibroblasts in the extracellular matrix [117,118].

## Materials and methods

### Cell lines and preparation

The cell lines used are the *Dictyostelium* wild-type AX3 and the mutants *scar*-null cells with a deletion of the *scrA* DDB_G0285253 gene [37], *pla2*-null with a deletion of the *plaA* DDB_G0278525 gene [38], *gc*-null with deletions of the *gcA* DDB_G0275009 gene encoding GCA and of the *sgcA* DDB_G0276269 gene encoding sGC [39], *gbpC*-null with a deletion of the *gbpC* DDB_G0291079 gene [41], *myoII*-null with a deletion of the *mhcA* DDB_G0286355 gene [119], *forAEH*-null with deletions of the *forA* DDB_G0279607 *forE* DDB_G0269626 gene and the *forH* DDB_G0285589 gene [47], *racE*-null cell with a deletion of the *racE* DDB_G0280975 gene [47], and the *lrrA*-null cells with a deletion of the *lrrA* DDB_G0294094 gene [50,51]. RapAG12V cells are AX3 wild-type that express the dominant active RapAG12V from an inducible promotor, and express the sensors Ral-GDS-GFP and cytosolic-RFP [120]. Cells were grown in HL5-C medium including glucose (ForMedium), containing the appropriate antibiotics for selection. Cells were collected and starved for 2–3 hours (unpolarized cells) or 5–6 hours (polarized). Cells were then harvested, suspended in 10 mM $KH_2PO_4$/$Na_2HPO_4$, pH 6.5 (PB), and used in experiments with microscopy recordings, using either phase contrast (Olympus Type CK40 with 20x objective and JVC CCD camera) or confocal laser scanning microscopy (Zeiss LSM800; 63x numerical aperture 1.4 objective). Unless mentioned otherwise images were recorded at a rate of 1 frame per second and an x,y-resolution of 245 nm for phase contrast and 198 nm for confocal microscopy, respectively. For chemotaxis, starved *Dictyostelium* cells were exposed to a stable cAMP gradient of 0.5 nM/μm as described [13]. For movement under agar, cells were covered with an approximately cubic (length 3 mm) block of 1.5% agar in PB [109]. The sources of the recordings are given in S1 Table.

### Pseudopod identification

The start and stop of pseudopod extension were identified as described previously [7,121], using the fully automated pseudopod tracking algorithm Quimp3, or a semi-automatic

pseudopod tracking macro for ImageJ. In Quimp3 the active contour algorithm identified the contour of a cell in all frames as polygons of about 120 nodes. A pseudopod is defined as an outward convex deformation of a spherical cell, and was identified with an algorithm using minimal parameters for convexity, area extension and extension period. Then the algorithm goes back and forward in time to identify the frame number (f) in which extension starts and stops, respectively. Finally, the central node of the convex area in these two frames identifies the tip of the pseudopod at the beginning and end of the extension period, respectively. The program reports on position of the tip at START and STOP, as well as on the area gain and the width of the pseudopod. In semi-automatic pseudopod tracking, the investigator identifies the start and final position of a pseudopod growth. The custom made macro exports position of the tip at START and STOP, and prints a hard-copy arrow on the relevant frames of the movie.

To select cells for pseudopod analysis, first the displacement during 15 min of all 20–30 cells in the field of observation was determined, then the 3–5 cells were selected that have a displacement closest to the mean displacement, and finally the 2–4 cells were selected that remain attached to the substratum during the entire movie. About 30 consecutive pseudopods were recorded from one cell, about three cells were recorded from one movie and at least two movies were recorded for each strain or condition (see S1 and S2 Tables for details). In this paper no discrimination is made between split pseudopods that start at the site of an extending pseudopod and de novo pseudopods that start at the cell body [7,9].

## Data analysis

For each pseudopod the position of the tip was identified in the frame immediately before extension started $(x_1,y_1,f_1)$ and again in the frame immediately after the extension terminated $(x_2,y_2,f_2)$.

These data provide an accurate estimate of the pseudopod size $S$, defined as the length of the vector connecting start and stop ($S = \sqrt{(x_2 - x_1)^2 + (y_2 - y_1)^2}$). For pseudopod growth time, please note that a pseudopod starts between frame $f_1$ and $f_1+1$, and stops between frame $f_2-1$ and $f_2$, and therefore the growth period of an extending pseudopod is on average one frame shorter than the difference ($f_2-f_1$). The start of all pseudopods was arbitrarily set at $f_1+1$ and the stop at $f_2$, and the growth time is given by $T = (f_2-f_1-1)^*$frame rate.

## Data fitting and model discrimination

Experimental data were analyzed using various models with different number of parameters. The parameter values of the model were varied to find the best fit between model and experimental data using the least residual sum of square (RSS) method. The goodness of the fit (95% confidence level) for linear models (e.g. Fig 2B) was obtained from the linear regression. For non-linear models (e.g. Fig 4E) the goodness of the fit was estimated by bootstrap analysis with random replacement of the data-set [122]. The Akaike Information Criterion (AIC) and the F-test [123–126] were used to select the optimal model, i.e. the model with the lowest number of parameters that fit the data significantly better than models with the same or less parameters:

$$AIC_c = 2p + Nln(RSS) + 2p(p + 1)/(n-p-1)$$

$$F = (N-p_2)/(p_2-p_1)*(RSS_2-RSS_1)/RSS_1$$

Where p is the number of parameters, N is the number of observations and the subscript 1 and 2 indicate the model with less and more parameters, respectively.

The model with the lowest AIC$_c$ value is identified as the preferred model; The F-test was used to calculate the significance (P-value) of the difference between two models.

## Supporting information

**S1 Table. Source of movies.**
(PDF)

**S2 Table. Overview of all data.**
(PDF)

**S1 Movie. RapAG12V phase contrast.**
(AVI)

**S2 Movie. RapAG12V Ral-GDS.**
(AVI)

**S1 File.**
(PDF)

## Acknowledgments

I am grateful to Ineke Keizer-Gunnink for the recording of many *Dictyostelium* mutants and to Arjan Kortholt for fruitful discussions.

## Author Contributions

**Conceptualization:** Peter J. M. van Haastert.

**Data curation:** Peter J. M. van Haastert.

**Formal analysis:** Peter J. M. van Haastert.

**Investigation:** Peter J. M. van Haastert.

**Methodology:** Peter J. M. van Haastert.

**Project administration:** Peter J. M. van Haastert.

**Resources:** Peter J. M. van Haastert.

**Validation:** Peter J. M. van Haastert.

**Visualization:** Peter J. M. van Haastert.

**Writing – original draft:** Peter J. M. van Haastert.

**Writing – review & editing:** Peter J. M. van Haastert.

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
