## [Decision Letter · Decision Letter 0]

24 Sep 2020

PONE-D-20-22458

Unified control of amoeboid pseudopod extension in multiple organisms by branched F-actin in the front and parallel F-actin/myosin in the cortex

PLOS ONE

Dear Dr. Van Haastert,

Thank you for submitting your manuscript to PLOS ONE.  Please accept my apologies for the fact that the review has taken such a long time. It  appeared quite challenging to find reviewers with time and expertise to carefully assess your manuscript. At the moment, your manuscript has been reviewed by two experts in the field, and you will find their extensive comments below. Both reviewers expressed enthusiasm in general, but also concerns about the definition of the parameters, which precluded a recommendation for publication of the manuscript in its current form. Therefore, we invite you to submit a revised version of the manuscript that addresses the points raised during the review process.

We look forward to receiving your revised manuscript.

Kind regards,

Mirjam M Zegers, Ph.D.

Academic Editor

PLOS ONE

Journal Requirements:

Reviewers' comments:

Reviewer's Responses to Questions

**Comments to the Author**

1. Is the manuscript technically sound, and do the data support the conclusions?

Reviewer #1: No

Reviewer #2: Partly

2. Has the statistical analysis been performed appropriately and rigorously? 

Reviewer #1: I Don't Know

Reviewer #2: Yes

3. Have the authors made all data underlying the findings in their manuscript fully available?

Reviewer #1: No

Reviewer #2: Yes

4. Is the manuscript presented in an intelligible fashion and written in standard English?

Reviewer #1: Yes

Reviewer #2: Yes

5. Review Comments to the Author

Reviewer #1: This seems to me to be potentially extremely important to the discourse on the field - quite separate from most of the rather shallow work on the nature of chemotaxis in recent years, and much more important than most.

However its excellence is really let down by a complete lack of definitions of terms and methods. I'm suggesting one more experiment, which is optional but which would really improve the interpretation. The rest is all about fuller explanation of the techniques and discussion of how they're different from anyone else's.

The principal one - what is a pseudopod? How is it being measured? Reading between the lines it's something to do with continuous protrusion at the edges, rather than the hyaline area that most authors have taken to be the pseudopod and which has some biological reason for being called a "pseudopod"*. Could the author please define in full how a pseudopod is defined, how it is measured, and what the limits are (eg if it's to do with continuous protrusion - is one frame of retraction enough to count as a stop? How much proportion of the cells arc must protrude to count as a pseudopod?). If it's different from the hyaline area please justify it and say what the differences are in a typical cell. I suggest a fig. 1A that shows a schematic of how the measurements are made, and Fig 1B a sequence of images showing how a pseudopod is called and what a typical stop looks like in real life.

This is quite important because the distribution of pseudopods described doesn't agree with for example, Andrew & Insall or Bosgraaf & van Haastert, as far as I can see. Are "new pseudopods" (this paper) the same as "split pseudopods", or would the splits just count as one continuous pseudopod (or were the cells so chosen that pseudopods come out sequentially rather than splitting?). Nice to see Swanson and Taylor being remembered.

It's particularly important because some of the measurements could be definitional. Pseudopod size, for example - I suspect the upper limit is defined by the cells, but the lower limit by the software or the experiments, so the range (which looks much smaller than I would have expected, for example in Fig 1c panel 3) might be completely defined by the definition of how small a pseudopod can be. The reader needs a clear definition of each measurement, how it's calculated, and how it relates to the DIC images shown.

Lesser, but important ones:

- what does "size" mean? It's a dimension but measured from what to what? I can't tell. Is it a vector or does it point perfectly along the cell axis?

- the effect of under-agar chemotaxis is nothing, I believe, to do with viscosity. Agar is a poro-elastic medium, but in any case I suspect the effects of agar are due to its weight (David Knecht measured this about 25 years ago in myosin KO papers) and the adhesive bonds between agarose and the plate, which just need to be broken once. Please rethink the text about it.

and -re mesenchymal stem cells - is this experiment being done fairly? Has the author compared Dictyostelium and mesenchymal cells filmed at the same frame rate and magnification? I have suspected the peak protrusion rate is similar but the mesenchymal cells protrude a little, then stop, then protrude a little more. Are the data good enough to rule this out? Often researchers take very long time steps between frames of epithelial cells, so you can't compare dynamics with the rapidly-moving cells. It would be great if that was not true in this paper.

Given these points sorted out - this will be a storming paper.

Reviewer #2: Summary: The manuscript by Dr. Van Haastert investigates the control of pseudopod dynamics in a variety of organisms and backgrounds. Using a combination of genetic perturbations and statistical analysis, they suggest that the formation of a pseudopod is a stochastic process that is strongly inhibited by the presence of other pseudopods, while the cessation of pseudopod extension is primarily controlled by intrinsic properties of the pseudopod itself. They then suggest a model based on their data in which the growth of the pseudopod is controlled by the ability of branched actin to push on the membrane. On the whole, the data presented here will be useful to the cell migration community both as a novel finding and as a resource. However, some additional detail is needed to ensure that the findings presented here are accurate. Once these concerns are addressed, I would recommend this article for publication

Major concerns:

One key observation the author reports is that both the START and STOP of pseudopod extension are stochastic, switch-like processes. This finding underlies the modeling assumptions made at several points in the paper. The main justification for this is found in Fig 1B, where the rate of extension increases rapidly from 0 to maximum levels immediately after START, and back to 0 immediately after STOP. To evaluate that the identification of the START and STOP points are accurate at the level of individual cells, it is important to see images of a single cell in the dataset for 1B with the START, STOP, and tip clearly identified. Also, the author should show some individual traces of extension rate in addition to average data over many cells. Finally, if START and STOP are switch-like processes, the rate of extension after START (but before STOP) and the extension before STOP (but after START) should be completely independent of time after START or before STOP, respectively. Is this true?

The quantification the author used for pseudopod size ( = √(2 − 1)2 + (2 − 1)2) actually only represents the length of the pseudopod. However, a pseudopod is a two-dimensional structure. The width of a pseudopod is also a key feature of the size of the pseudopod. For example, a long and narrow pseudopod may lead to different motion than a long and wide one. Can the author can still draw the similar major conclusions of paper if he quantifies the actual two-dimensional size instead of only the length of the pseudopod? For example, do Rap1G12V cells still have smaller pseudopods by this metric? Does STOP still primarily scale with size?

Similarly, many of the perturbations the author does alters overall cell morphology. RapG12V overexpression, for example, leads to flatter cells with a larger cross-sectional area. This flattening is due in part to the increase in PI3K activation and actin polymerization (eg Kortholt at al., MBOC 2010). This introduces an interesting paradox: while the size of what is designated a pseudopod is smaller in these cells, the exact same activities are affecting the shape of the entire cell boundary. Is it accurate to exclude these areas when analyzing pseudopod dynamics?

We know that cell behavior can vary from individual to individual in the same strain. The author has shown 39 cells for polarized WT Dictyostelium, which is good. However, especially given the fact that this paper focuses mainly on meta-analysis, the numbers of cells studied are not as satisfactory for the other cell strains: for instance, only 4 cells for gc‐null dicty, 4 cells for gbpC‐null dicty, 4 cells for myoII‐null dicty, 4 cells for Rap1G12V dicty, and 4 cells of B.d. chytrid. To address this concern, can the author show the same quantification data in Table S2 of each individual cell of the 4 cells of the above strains? If the workload is too heavy, can the author at least show the 4 individual cell data of one of the above 5 strains?

The author showed the data of polarized WT dicty in Fig. 2B and concluded that “inhibition A of START depends on the power of the number of extending pseudopods”, and generalized the conclusion to all 16 cell lines of different mutants and species with data in Fig. S1. This is a very interesting finding. In the Fig. S1, the author combined the 7 WT stains of different species (Dicty, neutrophils, mesenchymal cells and B.d. chytrid). Can the author can show data from individual cells of these 7 WT cells in another supplemental figure as he has shown in Fig. 2B?

Similar to the above point, for the data of Fig. 3B, can the author show some individual data of each strain in another supplemental figure?

Examining the quantitative descriptors of pseudopod dynamics over many genetic backgrounds is a very powerful way to examine the molecular mechanisms that govern this process. However, long-term perturbations to cytoskeletal components may affect cells in unforeseen ways. It might be useful to confirm some findings by transiently inhibiting cytoskeletal components. For example, does treatment of the cell with Formin inhibitor SMIFH2 lead to a decrease in A, similar to the forAEH-null cells?

Minor Concerns:

In reference to Fig. 4E, the author states, “With one parameter, only pseudopod stopping by size dependent inhibition (ks) can describe the observed data to some extent.” Given that the ks line does not intersect with the data at any point, it would useful to see the lines for kv and kt, on a separate graph if necessary, alone to evaluate this claim.

The author is primarily focused on the formation of Pseudopods, which are relatively small and transient. While they briefly mention Keratocytes, which have broad and long-lived actin fronts, they imply that these are of a different class and may be governed by different rules. However, the Devreotes lab has observed (Miao et al., NCB 2017) that Dictyostelium cells can be converted to a keratocyte-like mode of movement by simply altering the threshold at which the excitable network governing protrusion formation fires. If the model the author is proposing is correct, the counterforces which act on pseudopods must never be sufficient to prevent the insertion of new actin at the cell front in these cells. What changes would account for this change in behavior? Additionally, macropinocytotic cups form through many of the same mechanisms as pseudopodia. Do they follow the same rules as pseudopodia? If not, why?

Another important observation that the author makes is that the probability of Pseudopod extensions stopping grows linearly with time from START. This is based primarily on the linear fitting of data in Fig 4B and 4C. The data in 4B especially does not seem to fully fit a linear trend: How does the accuracy of a linear fit here compare with other models?

Fig 1B: are these 10 pseudopods all from one cell? Do different cells have different properties?

Fig 1D: does this data include all the 39 WT polarized dicty cells?

The methods part does not include cell culture conditions for Neutrophils, Mesenchymal cells, and B.d. chytrid. I know the author downloaded these movies for analysis from other studies. Maybe it is better to still indicate the cell conditions in the method part of this paper with the information from other papers?

There are a couple of minor formatting issues in References: for citation 15, the journal name “Nature reviews. Molecular cell biology. England" shall better be presented as “Nature Reviews Molecular Cell Biology”; for citation 58, the journal name “Nature cell biology. England” shall better be presented as “Nat Cell Biol” as shown in other citations.

Adjustments for clarity:

It would be helpful to note directly on the figures which type of Dicty cells (polarized, non-polarized) are being observed.

Fig 1C: “Duration” might be more clear than “time”

Fig 1D: “Number of simultaneous pseudopods” would be more clear

Fig 4E: Specify directly on the graph that the two ks + kv + kt lines are derived from different constant values

6. PLOS authors have the option to publish the peer review history of their article (what does this mean?). If published, this will include your full peer review and any attached files.

Reviewer #1: **Yes: **Robert Insall

Reviewer #2: No

---

## [Decision Letter · Decision Letter 1]

17 Nov 2020

PONE-D-20-22458R1

Unified control of amoeboid pseudopod extension in multiple organisms by branched F-actin in the front and parallel F-actin/myosin in the cortex

PLOS ONE

Dear Dr. Van Haastert,

Thank you for submitting your manuscript to PLOS ONE. My apologies, again, for the time it took to have it assessed by the two original reviewers. I am happy to let you know that based on their assessments, we feel that  in principle your manuscript can be accepted for publication. I would however recommend that you take the small suggested change of reviewer 2 in consideration and therefore invite you to submit a revised version of the manuscript that addresses this point.

We look forward to receiving your revised manuscript.

Kind regards,

Mirjam M Zegers, Ph.D.

Academic Editor

PLOS ONE

Reviewers' comments:

Reviewer's Responses to Questions

**Comments to the Author**

1. If the authors have adequately addressed your comments raised in a previous round of review and you feel that this manuscript is now acceptable for publication, you may indicate that here to bypass the “Comments to the Author” section, enter your conflict of interest statement in the “Confidential to Editor” section, and submit your "Accept" recommendation.

Reviewer #1: All comments have been addressed

Reviewer #2: (No Response)

2. Is the manuscript technically sound, and do the data support the conclusions?

Reviewer #1: Yes

Reviewer #2: Yes

3. Has the statistical analysis been performed appropriately and rigorously? 

Reviewer #1: Yes

Reviewer #2: Yes

4. Have the authors made all data underlying the findings in their manuscript fully available?

Reviewer #1: Yes

Reviewer #2: Yes

5. Is the manuscript presented in an intelligible fashion and written in standard English?

Reviewer #1: Yes

Reviewer #2: Yes

6. Review Comments to the Author

Reviewer #1: I'm happy with the changes made to this paper.

I'm also happy the formin inhibitor was not used, given the doubts about its specificity.

Delighted to see this paper published.

Reviewer #2: For the most part, I am satisfied with the response to my comments. However, I would still like to see some individual traces of vectorial rate vs. time in addition to the average trace in Figure 1B. This would help readers assess whether individual pseudopods behave as described by the bulk data. Unless I am mistaken, the author's previous papers do not contain such a plot. I feel this request is reasonable because it would not require any additional experiments and should be simple from an analysis standpoint.

7. PLOS authors have the option to publish the peer review history of their article (what does this mean?). If published, this will include your full peer review and any attached files.

Reviewer #1: **Yes: **Robert Insall

Reviewer #2: No

---

## [Editor Report · Decision Letter 2]

23 Nov 2020

Unified control of amoeboid pseudopod extension in multiple organisms by branched F-actin in the front and parallel F-actin/myosin in the cortex

PONE-D-20-22458R2

Dear Dr. Van Haastert,

We’re pleased to inform you that your manuscript has been judged scientifically suitable for publication and will be formally accepted for publication once it meets all outstanding technical requirements.

Kind regards,

Mirjam M Zegers, Ph.D.

Academic Editor

PLOS ONE
---

## [Editor Report · Acceptance letter]

27 Nov 2020

PONE-D-20-22458R2 

Unified control of amoeboid pseudopod extension in multiple organisms by branched F-actin in the front and parallel F-actin/myosin in the cortex 

Dear Dr. van Haastert:

I'm pleased to inform you that your manuscript has been deemed suitable for publication in PLOS ONE. Congratulations! Your manuscript is now with our production department. 

Kind regards, 

on behalf of

Dr. Mirjam M Zegers 

Academic Editor

PLOS ONE